# Nova: Generative Language Models for Assembly Code with Hierarchical Attention and Contrastive Learning

**Nan Jiang**
Purdue University
jiang719@purdue.edu

**Chengxiao Wang**
University of Illinois Urbana-Champaign
cw124@illinois.edu

**Kevin Liu**
Lynbrook High School
kevin.bx.liu@gmail.com

**Xiangzhe Xu**
Purdue University
xu1415@purdue.edu

**Lin Tan**
Purdue University
lintan@purdue.edu

**Xiangyu Zhang**
Purdue University
xyzhang@cs.purdue.edu

**Petr Babkin**
J.P. Morgan AI Research
petr.babkin@jpmorgan.com

## Abstract

Binary code analysis is the foundation of crucial tasks in the security domain; thus building effective binary analysis techniques is more important than ever. Large language models (LLMs) although have brought impressive improvement to source code tasks, do not directly generalize to assembly code due to the unique challenges of assembly: (1) the low information density of assembly and (2) the diverse optimizations in assembly code. To overcome these challenges, this work proposes a *hierarchical attention* mechanism that builds attention summaries to capture the semantics more effectively, and designs *contrastive learning objectives* to train LLMs to learn assembly optimization. Equipped with these techniques, this work develops *Nova*, a generative LLM for assembly code. Nova outperforms existing techniques on binary code decompilation by up to $14.84 - 21.58\%$ (absolute percentage point improvement) higher Pass@1 and Pass@10, and outperforms the latest binary code similarity detection techniques by up to 6.17% Recall@1, showing promising abilities on both assembly generation and understanding tasks.

## 1 Introduction

Binary code plays an irreplaceable role in the security domain, being the foundation of crucial tasks including vulnerability detection (Güler et al., 2019; Duan et al., 2020; Chen et al., 2022b), malware detection (Spensky et al., 2016; Aonzo et al., 2023; Xu et al., 2014), binary recovery (Su et al., 2024; Zhang et al., 2021; Chen et al., 2022c), and legacy software maintenance (Carbone et al., 2009; Carlini et al., 2015; Martin et al., 2010). For example, when performing tasks such as identifying attacks and malware, security analysts often only have access to assembly, i.e., the human-readable representation of binary code, which is extremely difficult to understand (Su et al., 2024; Zhang et al., 2021; Chen et al., 2022c). Thus, combined with the increasing sophistication of cybercrime that poses significant threats worldwide (e.g., cybercrime is predicted to cost the world \$10.5 trillion annually by 2025 (Sausalito, 2020)), effective binary analysis techniques are in high demand.

Figure 1: Example that shows the semantics and diverse optimizations of assembly code.

Large language models pre-trained on source code have brought improvement in various software development domains (Chen et al., 2022a; Liu et al., 2023a; Chen et al., 2023; Le et al., 2022; Jiang et al., 2023; Xia et al., 2023). However, these LLMs are not designed for or trained with assembly corpus, not achieving their full potential on binary code analysis tasks such as binary code similarity (Wang et al., 2022; Xu et al., 2023a), malware detection (Su et al., 2024), and binary code decompilation (Tan et al., 2024; Armengol-Estapé et al., 2024; Hosseini & Dolan-Gavitt, 2022).

Existing work applying LLMs on assembly code mainly piggybacks on encoder-style LLMs (Wang et al., 2022; Su et al., 2024; Xu et al., 2023a), unable to benefit from the more extensive pre-training, updated architectures, scaling of state-of-the-art generative LLMs. Other work using generative LLMs for decompilation shows a low unit test passing rate of the decompiled programs (Tan et al., 2024; Armengol-Estapé et al., 2024).

The challenges of leveraging generative LLMs for assembly code are twofold. First, compared to source code, assembly code has a *lower information density*. A short source-code sequence maps to an assembly-code sequence that is often several times longer. Thus, assembly semantics span across a *long sequence of tokens*. Figure 1 (a) shows an example of a source code function that compares two integers, while Figure 1 (b) shows its corresponding assembly code optimized with `00` flag. In the `00`-optimized assembly code, the five instructions from `10: mov -0x8(%rbp),%rax` to `1c: cmp %eax,%edx` perform the checking whether the value of x is smaller than the value of y (correspond to `if (*(int*)x < *(int*)y)` in the source code). A single assembly instruction alone represents little meaningful semantics in the source code. It is the combinations of *many instructions* and the *dependencies* between them represent the semantics. Such combinations of instructions are long, which is hard for LLMs to learn.

Second, assembly code is diverse due to compiler optimization. The assembly code of the same source code function looks dramatically different with different compiler optimization. Figure 1 (c) shows the assembly of the same function compiled with `01` and `00` flags, which consists of a significantly different set of instructions. Such syntax diversity is hard for LLMs to learn, preventing LLMs from obtaining consistently good performances on differently optimized assembly code.

In this work, we develop Nova, a generative foundation LLM pre-trained for assembly code with two key novelties. First, to address the low-information-density and long-sequence challenge, we design a hierarchical self-attention, which contains three categories of attention at different levels of granularity: intra-instruction attention, preceding-instruction attention, and inter-instruction attention. The key insight is to build *attention summaries*, i.e., we create per-statement attention *labels*, which act as the summary of a statement. We then use preceding-instruction attention to capture semantics between a token and its preceding instruction label and use inter-instruction attention for long dependencies. Besides, we design *functionality contrastive learning* and *optimization contrastive learning* objectives to train Nova to learn the semantics behind the diverse syntax of assembly.

This work makes the following contributions:

- We propose a novel hierarchical attention mechanism that captures the assembly's low-density semantics at three granularity levels.
- We design contrastive learning objectives to train LLMs to learn assembly with diverse optimizations and encode assembly more efficiently.
- We develop Nova, a generative foundation LLM with hierarchical attention and contrastive learning for assembly. Nova outperforms state-of-the-art on binary decompilation by up to 14.84 – 21.58% higher Pass@1 and Pass@10, and on binary similarity detection by up to 6.17% Recall@1.
- Availability: we release Nova models at `https://huggingface.co/lt-asset/nova-1.3b` and `https://huggingface.co/lt-asset/nova-6.7b`

## 2 RELATED WORK

### 2.1 BINARY MODELS

Machine learning models are widely used in binary program analysis tasks. However, these models are typically designed for specific tasks such as binary code similarity detection (Pei et al., 2020; Xu et al., 2023a; Wang et al., 2022; Xu et al., 2017; Ding et al., 2019), variable name prediction (Chen et al., 2022c; Xu et al., 2023b; Zhang et al., 2021; He et al., 2018; Lacomis et al., 2020), binary

code type inference (Pei et al., 2021), and so on Chen et al. (2022d); Liu et al. (2023b); Hosseini & Dolan-Gavitt (2022).

Recent techniques have started to pre-train foundation LLMs for binaries. CodeArt (Su et al., 2024) pre-trains encoder-style LLMs with a regularized attention design to better encode assembly code semantics. SLaDe (Armengol-Estapé et al., 2024) trains BART (Lewis et al., 2019) models on assembly. Meta LLMCompiler (Cummins et al., 2024) train CodeLlama models on LLVM IR to optimize binary code. LLM4Decompile (Tan et al., 2024) trains DeepSeekCoder with assembly for binary code decompilation. However, CodeArt does not generalize to generation tasks due to its encoder architecture. LLMCompiler trained on LLVM IR cannot be effectively transferred to assembly code. SLaDe and LLM4Decompile are limited in performance due to a lack of special designs for assembly. In contrast, Nova proposes hierarchical attention and contrastive learning objectives, outperforming existing techniques on both understanding (binary code similarity detection) and generation (binary code decompilation) tasks.

## 2.2 LARGE SOURCE-CODE MODELS

LLMs demonstrate promising results on many code-related tasks, such as code generation (Chen et al., 2022a; Liu et al., 2023a; Chen et al., 2023; Le et al., 2022; Yue et al., 2021; Chen et al., 2021; Nijkamp et al., 2022; Fried et al., 2023; Rozière et al., 2023; Guo et al., 2024; Lozhkov et al., 2024; Hui et al., 2024), bug fixing (Jiang et al., 2023; Xia et al., 2023) and vulnerability fixing (Wu et al., 2023; Steenhoek et al., 2023; He & Vechev, 2023). The advances in using LLMs are attributed to the knowledge learned from massive source code and natural language text in their training datasets (Touvron et al., 2023; OpenAI, 2023). Nova is designed and trained for assembly, which has unique challenges such as low information density and diverse optimization.

## 2.3 ATTENTION MECHANISM

Standard self-attention is widely used in transformer architecture (Vaswani et al., 2017) to capture soft dependencies between tokens in the input. Many special attention mechanisms have been designed for better learning in various scenarios (Yang et al., 2016; Huang et al., 2024). LongCoder (Guo et al., 2023) combines window attention and global attention (attention sink (Xiao et al., 2024)) to handle long input of source code. We have shown that LongCoder's window attention is less effective than Nova's on assembly code. CAST (Shi et al., 2021) is a new neural architecture that splits the abstract syntax tree (AST) of source code into subtrees, encodes the subtrees, and aggregates to the final encoding. PA-former (Chai & Li, 2024) is a new neural architecture that constructs source code as pyramid input based on their AST structure and contains a pyramid attention mechanism to calculate the features in a hierarchical way. HierarchyNet (Nguyen et al., 2023) is a neural architecture that considers source code AST, data flow, and control flow graphs. Similarly, it cannot be applied to assembly code. Different from CAST, PA-former, and HierarchiyNet, Nova's attention design is for assembly code, is more lightweight and can be plugged into any pre-trained generative LLM.

## 3 APPROACH

Figure 2 presents the overall approach of Nova. We build Nova on top of foundation models for source code (Guo et al., 2024) to utilize their source code and natural language generation ability. We first collect large assembly corpora (Section 3.1). Section 3.2 describes Nova's hierarchical attention design. With the collected assembly corpora, we then pretrain Nova with language modeling and contrastive learning objectives (Section 3.3). Then, we fine-tune Nova on two important downstream tasks, binary code decompilation, and binary code similarity detection (Sections 3.4 and 3.5), to prove Nova's effectiveness and benefits to the binary research domain.

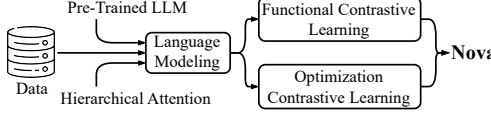

Figure 2: Overview of developing Nova

Table 1: Statistics (number of C and X86-64 assembly functions) of the pre-training datasets.

| Origin | C Functions | O0 | O1 | O2 | O3 | Total |
|---|---|---|---|---|---|---|
| AnghaBench | 757.1K | 743.1K | 726.4K | 718.7K | 717.8K | 3.7M |
| The-Stack | 138.8K | 125.1K | 119.7K | 116.9K | 108.8K | 609.3K |

## 3.1 DATA COLLECTION

In this paper, we focus on X86-64 assembly functions for C programs. Yet, Nova's approach is generalizable to other assembly languages such as ARM assembly.

We derive our X86-64 assembly functions dataset from two source code corpora: C functions in The-Stack (Li et al., 2023) and AnghaBench (da Silva et al., 2021). We compile the C programs into executables using gcc with different optimization levels (i.e., O0, O1, O2 and O3), strip the executables to remove debug information, and disassemble them into X86-64 assembly code using objdump. We treat every function as a separate data point. Table 1 shows the number of C functions in the two original datasets, and the number of X86-64 assembly functions we collected from them.

We perform certain normalization on the assembly functions: (1) removing all the "%" and comments, (2) adding whitespace around ",", "(", ")", (3) converting all the hexadecimal numbers to decimal numbers, and (4) replacing the address of each instruction with special labels (e.g., replacing "0" and "4" in Figure 1 (b) with "[INST-1]" and "[INST-2]") placing at the end of each instruction. More details are in Appendix A.1.

## 3.2 HIERARCHICAL SELF-ATTENTION

Nova uses hierarchical self-attention that is specially designed to learn the *low-information-density* semantics in the *long* sequence of assembly code. Specifically, Nova learns the assembly code in an hierarchical way by providing a modified attention mask. Different from standard token-level attentions (Vaswani et al., 2017; Radford & Narasimhan, 2018; Radford et al., 2019; Brown et al., 2020), our hierarchical self-attention contains three categories at different levels of granularity.

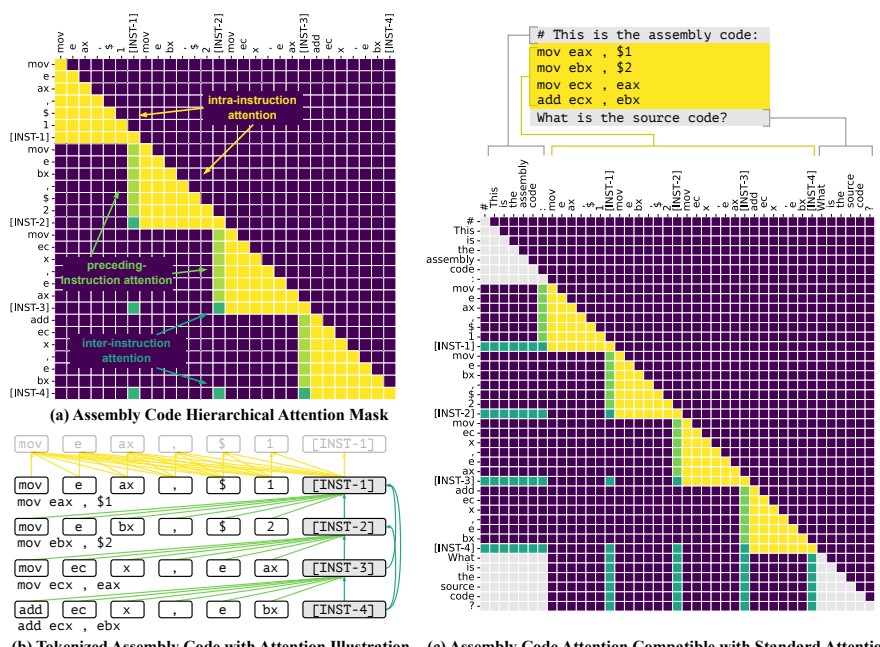

(a) Assembly Code Hierarchical Attention Mask

(b) Tokenized Assembly Code with Attention Illustration     (c) Assembly Code Attention Compatible with Standard Attention

Figure 3: Design of Nova's hierarchical attention for assembly code

**(1) Intra-Instruction Attention:** Due to the low information density in assembly, intra-instruction attention is designed to capture the summary of every instruction, which is the standard causal attention but limited to tokens of each instruction (the yellow part in Figures 3) (a) and (b). Tokens in different instructions have no attention weights. The "[INST]" label at the end of the instruction has attention to all the tokens in the instruction and thus captures the semantics of the entire instruction (e.g., "[INST-1]" captures the semantics of "mov eax, $1").

**(2) Preceding-Instruction Attention:** In addition to the local semantics of each instruction, the use of assembly instructions (such as the choice of registers) depends on the context. For example, after

the first instruction "`mov eax, $1`", the second instruction should not reuse "`eax`" to store another value "`$2`" immediately. To capture such context, the preceding-instruction attention enables each token in an instruction to have attention to the "`[INST]`" label of the preceding instruction (the light green part in Figures 3 (a) and (b)).

**(3) Inter-Instruction Attention:** To understand function semantics (i.e., functionality), which lies in the dependencies across different instructions, the inter-instruction attention is designed to let the "`[INST]`" label of each instruction have attention to all the labels of previous instructions. For example, "`[INST-4]`" has attention to "`[INST-1]`", "`[INST-2]`", and "`[INST-3]`" (the dark green part in Figures 3 (a) and (b)). The inter-instruction attention is only enabled for "`[INST]`" labels, as they represent the semantics of each instruction.

To sum up, the hierarchical self-attention splits assembly code semantics into three levels: intra-instruction attention captures instruction summaries, preceding-instruction attention provides context from the preceding instruction, and inter-instruction attention models long dependencies across instructions using "`[INST]`" tokens that contain the instruction summary. Figure 3 (c) highlights the compatibility of Nova's hierarchical attention with standard self-attention for text and source code. Leveraging the proven performance of standard self-attention in existing LLMs, we retain the causal attention mask within and across chunks of text or source code (shown in light grey in Figure 3 (c)). Cross-attention between text, source code, and assembly is restricted to "`[INST]`" tokens, which encapsulate assembly instruction summaries.

### 3.3 Contrastive Learning

The syntax gap between assembly code and source code, and syntax diversity between differently-optimized assembly code make LLMs struggle to distinguish the semantics behind the syntax. Nova adopts contrastive learning technique (Gao et al., 2021) during pre-training to train LLMs to encode assembly code meaningfully w.r.t semantics. The standard pre-training objective is language modeling by minimizing the negative likelihood of code in the pre-training corpus (Radford & Narasimhan, 2018), notated as $L_{lm}$. In addition, Nova is pre-trained with two new objectives, $L_{fcl}$ for functionality contrastive learning and $L_{ocl}$ for optimization contrastive learning.

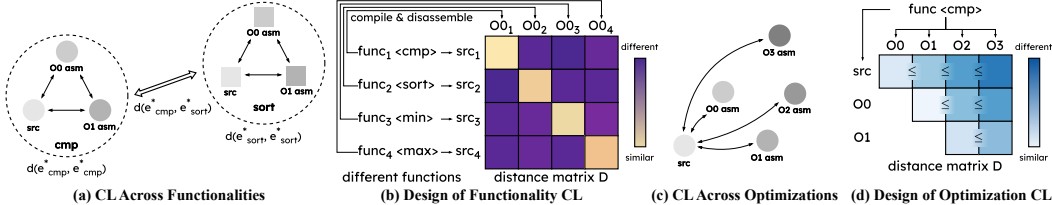

Figure 4: Design of functionality and optimization contrastive learning (CL). "`asm`" denotes assembly.

**Functionality CL:** Functionality CL trains Nova to focus more on the functionalities of assembly code rather than the syntax. Code with the same functionality (assemblies from the same source code), should be encoded closer in the latent space. For instance, in Figure 4 (a), embeddings of source and assembly code of function "`cmp`" are closer to each other, and the same for function "`sort`".

Nova is designed and implemented on decoder-only generative LLMs, and we refer the hidden states from the last transformer layer as embedding. For source code, we use the average of each token's embedding as the source code function's embedding. For assembly, we use the average of all the "`[INST]`" tokens' embedding as the embedding of the assembly function, as each "`[INST]`" token is supposed to capture the semantics of that instruction by the design of our hierarchical self-attention.

Let $e_f^s$ be the embedding of function $f$ in $s$ form ($s = -1$ for source code, and $s \in [0, 1, 2, 3]$ for `O0` to `O3` optimized assembly). For simplicity, let $S = [-1, 0, 1, 2, 3]$ be the domain of $s$. Functionality CL optimizes Nova's embeddings to satisfy the constraint:

$$\forall f_i \in F,\ \max_{s,t \in S}(d(e_{f_i}^s, e_{f_i}^t)) < \min_{\substack{s,t \in S \\ f_j \neq f_i \in F}} (d(e_{f_i}^s, e_{f_j}^t))$$

, where $d$ calculates the $l_2$ distance between two embeddings and $F$ is the full set of functions in the training corpus.

The embeddings of a batch of functions, each represented in two different forms, can be optimized to satisfy these constraints. For the example in Figure 4 (b), there are two forms (source code and O0 assembly) of four functions. Once Nova encodes the batch of source code and assembly functions, we calculate the distance matrix $\{D_{ij}\}_{f_i, f_j \in F} = \{d(e^s_{f_i}, e^t_{f_j})\}$, and minimize the loss:

$$L_{fcl} = -log \sum_{s,t \in S} \sum_{f_i \in F} \left( 1 - \frac{exp\left(d(e^s_{f_i}, e^t_{f_i})\right)}{\sum_{f_j \in F} exp\left(d(e^s_{f_i}, e^t_{f_j})\right)} \right)$$

This objective minimizes the distance between the embeddings of the same function, which is the diagonal in the distance matrix. In practice, this loss is computed and back-propagated for each batch of functions. Given a batch size of 64, $F$ represents a set of 64 unique functions in the batch.

**Optimization CL:** LLMs can be confused if being asked to directly connect a source code function to its O3-optimized assembly, due to their dramatically different syntax. Such a huge gap can be filled by learning how the source code is transformed to O0, O1, O2 and eventually to O3 assembly, as the optimization levels are *ordered*. Higher-level optimization applies a super-set of optimization rules compared to lower-level optimization.

Nova learns such order with the optimization CL objective, encoding differently-optimized assembly code orderly. Optimization CL optimizes Nova with the constraint: the more optimizations applied, the larger the difference between embeddings of optimized and unoptimized code. For instance, Figure 4 (c) and (d) illustrate that for the same function "cmp", the distance between source code and assembly increases when the optimization level increases. Formally, optimization CL minimizes the following loss:

$$L_{ocl} = \sum_{f \in F} \sum_{s < t_1 < t_2 \in S} max\left(0, d(e^s_f, e^{t_1}_f) - d(e^s_f, e^{t_2}_f)\right)$$

Overall, the final training loss combines the three: $L = L_{lm} + \lambda(L_{fcl} + L_{ocl})$, where $\lambda$ is set to 0.1 to balance the losses in this work.

## 3.4 TASK 1: BINARY CODE DECOMPILATION

Binary code decompilation (BCD) helps developers to understand binary code by recovering binary code into more readable high-level source code (e.g., C programs) (Fu et al., 2019; Liang et al., 2021; Armengol-Estapé et al., 2024; Tan et al., 2024). The input to the model for BCD is formatted as an instruction prompt (notated by $\mathbf{p}$): # This is the assembly code with {opt} optimization: {asm}, where "opt" is the optimization-level applied to the assembly and "**asm**" is the assembly code to decompile. Nova is fine-tuned to generate the expected source code function **src** following the instruction prompt. The fine-tuning objective is minimizing the loss: $L_{bcd} = -log\, P(\mathbf{src}|\mathbf{p})$.

## 3.5 TASK 2: BINARY CODE SIMILARITY DETECTION

Binary code similarity detection (BCSD) aims to measure the similarity between two binary code snippets (Wang et al., 2022; Su et al., 2024), which is the foundation of various applications such as plagiarism detection (Luo et al., 2014; Sæbjørnsen et al., 2009) and vulnerability detection (David & Yahav, 2014; David et al., 2018; 2017; 2016).

A widely used setting is taking a query assembly of the function $f^q$ that is compiled with one optimization level (denoted by $s$), and a pool of candidate assembly of $K$ (e.g., 50, 100, etc.) functions (notated by $f^p_i$, $1 \le i \le K$) compiled with a different optimization level (denoted by $t \ne s$). There exists a unique candidate assembly coming from the same source code as the query ($\exists! 1 \le i \le K, f^p_i = f^q$, called the positive candidate). Nova is fine-tuned to encode these binaries, so that the positive candidate has the highest similarity with the query assembly among the pool. The learning objective is as follows:

$$L_{BCSD} = -log \sum_{1 \le i \le K}^{f^q := f_i^p} \left( 1 - \frac{exp\left(d(e_{f^q}^s, e_{f_i^p}^t)\right)}{\sum_{1 \le j \le K} exp\left(d(e_{f^q}^s, e_{f_j^p}^t)\right)} \right)$$

We follow previous work (Su et al., 2024) to let $s$ be `00`-assembly and $t$ be `03`-assembly, which is the hardest setting.

## 4 EXPERIMENTAL SETUP

This section describes the setup of pre-training and fine-tuning of Nova, as well as the existing baselines we compare Nova with, and the evaluation metrics we used in the two downstream tasks. Appendix A.2 contains additional details such as training hyper-parameters, and evaluation setup.

### 4.1 PRE-TRAINING

We use the C and X86-64 assembly functions collected from AnghaBench and The-Stack for pre-training. We pre-train Nova starting from DeepSeek-Coder (Guo et al., 2024), and the hierarchical attention is applied on half of the attention heads to balance between its effectiveness and the existing knowledge in the standard attention layers (Justified in Appendix A.3). Nova is pre-trained with language modeling for one epoch, followed by contrastive learning objectives for another epoch.

### 4.2 FINE-TUNING FOR BINARY CODE DECOMPILATION

**Training Data:** We sample (due to computation resource limitation) 2.16M assembly-to-source-code pairs (0.338B tokens) from the pre-training corpus to build the BCD fine-tuning data.

**Test Data:** We use HumanEval-Decompile (Tan et al., 2024) as the test benchmark, which was not used in training. HumanEval-Decompile is derived from the C language adaptation of the HumanEval (Chen et al., 2021) benchmark and contains 164 C functions, each compiled with `00 − 03` optimization flags and disassembled into X86-64 assembly.

**Baselines:** Nova is compared with existing SOTA LLM4Decompile (Tan et al., 2024), Meta LLM-Compiler (Cummins et al., 2024), other open-sourced general code LLMs (Rozière et al., 2023; Lozhkov et al., 2024; Guo et al., 2024; Hui et al., 2024; Mishra et al., 2024; Guo et al., 2023), and commercial LLMs GPT-3.5-Turbo and GPT-4o). LLM4Decompile trains DeepSeekCoder using the same AnghaBench corpus for binary decompilation. Meta LLMCompiler trains CodeLlama models using LLVM IRs, X86, and ARM assembly code to optimize and translate binary code.

**Evaluation:** We let each model sample 20 decompilations per assembly function, using the temperature of 0.2 and `top_p` of 0.95 (Chen et al., 2021). Except for LLM4Decompile and Nova that are fine-tuned for binary code decompilation, we provide all other baselines with three-shot examples for few-shot learning (Brown et al., 2020). The decompilations are executed with the test cases to verify the functional correctness. Finally, Pass@1 and Pass@10 (Chen et al., 2021) are reported.

### 4.3 FINE-TUNING FOR BINARY CODE SIMILARITY DETECTION

**Training Data:** To compare Nova with existing works on BCSD fairly (Wang et al., 2022; Su et al., 2024), we use the same dataset, BinaryCorp-3M (Wang et al., 2022), as the fine-tuning data for BCSD, which contains the `00` and `03` assembly of 224,606 functions.

**Test Data:** Following existing work (Su et al., 2024; Xu et al., 2023a), we use real-world benchmarks, Binutils, Curl, ImageMagick, SQLite, OpenSSL, and Putty, as the test benchmarks, which are nonexistent in the training data.

**Baselines:** Nova is compared with jTrans (Wang et al., 2022), DiEmph (Xu et al., 2023a) and CodeArt (Su et al., 2024). jTrans is a Transformer (Vaswani et al., 2017) encoder trained on binaries with masked token prediction and jump target prediction tasks. DiEmph uses an instruction deemphasis technique to prevent the model from learning instruction distribution biases introduced by

compilers. CodeArt proposes a regularized attention mask for encoder models to capture instructional semantics and data dependencies.

**Evaluation:** We randomly sample $K = 50, 100, 200, 500$ source code functions from each project, compile them into binaries with `O0` and `O3` optimization flags, and disassemble them into X86-64 assemblies. BCSD techniques encode these assemblies into embeddings (Nova uses the average last-layer hidden states of all the "`[INST]`" tokens in an assembly as its embedding). Then each `O0` assembly is used as the query to calculate their similarity with the $K$ `O3` candidate assemblies. Metric Recall@1 is reported as the ratio of queries for which the candidate from the same source code has the highest similarity among all the candidates.

# 5 RESULTS

## 5.1 BINARY CODE DECOMPILATION

**Comparison with SOTA Techniques:** Table 2 shows the Pass@1 and Pass@10 of the decompiled code from assemblies on HumanEval-Decompile. The results are grouped by optimization level (i.e., the benchmark contains 164 assemblies of each optimization level), and the average is also reported.

Table 2: Nova's Pass@K and comparison with existing techniques on HumanEval-Decompile.

| Techniques | Pass@1 | | | | | Pass@10 | | | | |
|---|---|---|---|---|---|---|---|---|---|---|
| | O0 | O1 | O2 | O3 | Avg. | O0 | O1 | O2 | O3 | Avg. |
| CodeLlama-7B | 6.95 | 3.81 | 4.54 | 3.78 | 4.77 | 8.53 | 5.97 | 7.34 | 5.17 | 6.75 |
| StarCoder2-7B | 6.31 | 4.33 | 5.64 | 5.95 | 5.56 | 8.77 | 5.18 | 6.09 | 7.17 | 6.80 |
| DeepSeekCoder-7B | 9.63 | 7.56 | 7.41 | 6.68 | 7.82 | 13.60 | 11.38 | 11.52 | 9.44 | 11.49 |
| Qwen-2.5-Coder-7B | 4.76 | 5.79 | 5.58 | 5.27 | 5.35 | 6.34 | 7.69 | 6.56 | 5.79 | 6.60 |
| LLMCompiler-7B | 5.95 | 5.85 | 5.55 | 5.82 | 5.79 | 7.01 | 7.31 | 7.47 | 7.01 | 7.20 |
| GPT-3.5-Turbo | 7.41 | 6.13 | 4.33 | 3.90 | 5.44 | 9.56 | 8.38 | 6.23 | 5.12 | 7.32 |
| GPT-4o | 21.34 | 18.29 | 14.48 | 13.05 | 16.79 | 29.94 | 26.74 | 21.42 | 19.88 | 24.50 |
| LLM4Decompile-1.3B | 15.30 | 8.26 | 9.36 | 8.38 | 10.33 | 21.79 | 15.23 | 16.17 | 13.70 | 16.72 |
| **Nova-1.3B** | **37.53** | **21.71** | **22.68** | **18.75** | **25.17** | **49.38** | **34.84** | **36.95** | **32.03** | **38.30** |
| LLM4Decompile-6.7B | 29.97 | 19.05 | 20.46 | 18.32 | 21.95 | 40.40 | 27.75 | 28.85 | 28.51 | 31.38 |
| **Nova-6.7B** | **48.78** | **30.58** | **30.85** | **27.23** | **34.36** | **57.47** | **47.45** | **43.03** | **39.68** | **46.91** |

*Overall, Nova's Pass@1 and Pass@10 are higher than all SOTA binary decompilation techniques and general LLMs with even smaller model sizes.* Specifically, for each optimization level, Nova consistently decompiles more assemblies into source code correctly than the rest of the compared techniques. Note that Meta LLMCompiler is mainly designed for LLVM IR optimization, and thus is still incapable of assembly code decompilation.

With the same model size, Nova-1.3B outperforms LLM4Decompile-1.3B with a 14.84% higher averaged Pass@1, and a 21.58% higher Pass@10. Nova-6.7B outperforms LLM4Decompile-6.7B with a 12.41% higher averaged Pass@1, and a 15.53% higher Pass@10. When compared with GPT-4o, an order of magnitude larger model, Nova-1.3B produces an 8.38% higher Pass@1 and 13.80% higher Pass@10. Examples of Nova's correct decompilation are provided in Appendix A.5.

**Comparison with Techniques Handling Long Input:** Nova's hierarchical attention design targets to address the low information density and long input challenge of assembly code. There are other techniques that handles long input challenges in text and source code, with Granite-3B-Code-Base-128K and LongCoder being the most related ones.

Granite trains LLM on repository-level long inputs, which is an orthogonal approach with Nova's approach (hierarchical attention and contrastive learning). We train Granite-3B-Code-128K with Nova's approach, and Table 3 shows that Nova's approach brings improvement to Granite over standard fine-tuning even if it has already been trained with long code data.

LongCoder combines window attention and global attention to learn long code input. We compare LongCoder's attention design with Nova's hierarchical attention design by replacing the hierarchical attention of Nova-1.3B with LongCoder's attention design. Table 4 shows that Nova's hierarchical attention is more effective in learning assembly code. Nova's attention design considers the instruction-

Table 3: Nova's approach brings improvement to LLM that has been trained with long input data.

| Techniques | Pass@1 | | | | | Pass@10 | | | | |
|---|---|---|---|---|---|---|---|---|---|---|
| | O0 | O1 | O2 | O3 | Avg. | O0 | O1 | O2 | O3 | Avg. |
| Granite (3B-Code-128K) | 5.91 | 3.78 | 5.09 | 5.52 | 5.08 | 8.19 | 5.16 | 6.56 | 7.15 | 6.76 |
| Granite + Standard Fine-Tuning | 20.88 | 13.54 | 11.37 | 10.09 | 13.97 | 30.05 | 19.77 | 18.31 | 15.77 | 20.98 |
| **Granite + Nova's Approaches** | **31.04** | **14.57** | **14.70** | **13.66** | **18.49** | **39.57** | **21.23** | **21.77** | **19.82** | **25.60** |

Table 4: Nova's hierarchical attention is more effective on assembly code.

| Techniques | Pass@1 | | | | | Pass@10 | | | | |
|---|---|---|---|---|---|---|---|---|---|---|
| | O0 | O1 | O2 | O3 | Avg. | O0 | O1 | O2 | O3 | Avg. |
| Nova-1.3B (using LongCoder's Attention) | 34.59 | 19.07 | 19.72 | 17.34 | 22.68 | 42.19 | 32.37 | 32.86 | 29.04 | 34.12 |
| **Nova-1.3B** | **37.53** | **21.71** | **22.68** | **18.75** | **25.17** | **49.38** | **34.84** | **36.95** | **32.03** | **38.30** |

level local semantics and dependencies between different instructions, which fits better than fix-sized window attention to assembly code.

**Ablation Study:** We conduct an ablation study by comparing Nova-1.3B with the following models to show the effectiveness of contrastive learning objectives and hierarchical attention design:

- $\text{Nova}_{-CL-HA}$: Removing contrastive learning and hierarchical self-attention. This is simply training DeepSeekCoder-1.3B on the assembly corpus using language modeling. This can be viewed as our reproduction (retrain) of LLM4Decompile-1.3B using the same amount of data.
- $\text{Nova}_{-HA}$: Removing the hierarchical self-attention, training DeepSeekCoder-1.3B on the assembly corpus using both the language modeling and contrastive learning objectives.

Table 5 shows the results of the ablation study. $\text{Nova}_{-CL-HA}$ produces an average Pass@1 of 15.78% and Pass@10 of 27.41%. With additional contrastive learning objectives, $\text{Nova}_{-HA}$ improves the Pass@1 on all optimization levels over $\text{Nova}_{-CL-HA}$, showing a higher averaged Pass@1 and Pass@10. Further applying the hierarchical self-attention on $\text{Nova}_{-HA}$ boosts the overall Pass@1 from 21.86% to 25.17%, and Pass@10 from 35.25% to 38.30%.

## 5.2 Binary Code Similarity Detection

Tables 6, 7, 8 and 9 show the Recall@1 of Nova and existing BCSD techniques with pool size $K$ of 50, 100, 200 and 500 on the six benchmarks. Underline indicates the best in each benchmark, while wave denotes the tied best (we only mark Nova-1.3B for clearer illustration).

Overall, Tables 6, 7, 8 and 9 show that *on average, Nova-1.3B and Nova-6.7B achieve the highest Recall@1 (in **bold**) under all four settings of $K$.* Nova-6.7B further outperforms Nova-1.3B and achieves the highest averaged Recall@1 under all four settings, ranking the ground-truth of 5%, 2%, 4%, and 3% more queries the most similar correspondingly compared to CodeArt. Nova-1.3B consistently outperforms existing techniques with higher Recall@1 when $K$ is 50, 100, and 200, meaning it correctly ranks ground-truth of 3%, 1%, and 2% more queries as the most similar. Under the setting of $K = 500$, Nova-1.3B ties with CodeArt with the same highest Recall@1. When looking into each individual benchmark, Nova-1.3B always wins on the most benchmarks under different settings of pool size $K$. For instance, Nova-1.3B wins on four benchmarks while DiEmph only wins on SQLite when $K = 50$. We also conduct an ablation study on BCSD in Appendix A.6.

## 5.3 Analytic Experiments: How are Nova's embeddings better?

We use the widely-used t-SNE (van der Maaten & Hinton, 2008) to analyze and visualize high-dimensional embeddings. We randomly sample seven coding problems from HumenEval-Decompile (`task_id` 19, 32, 34, 63, 119, 128, 143), encode the `O0 − O3` assemblies by $\text{Nova}_{-CL-HA}$ and Nova-1.3B. Figure 5 shows the embeddings that are visualized under the first two principal components. Each color represents one task, and `O0 − O3` assemblies are marked by $\bigcirc$, $\bigtriangledown$, $\triangle$, and $\square$.

Compared with $\text{Nova}_{-CL-HA}$ (Figure 5 (a)), $\text{Nova}_{-HA}$ (Figure 5 (b)) including contrastive learning objectives in the pre-training, can separate the embeddings of assemblies with different functionalities better. $\text{Nova}_{-HA}$ clearly encode "Task 143" (orange points) away from the others. Yet, Nova's

Table 5: Ablation study of Nova-1.3B on HumanEval-Decompile.

| Techniques | Pass@1 | | | | | Pass@10 | | | | |
|---|---|---|---|---|---|---|---|---|---|---|
| | O0 | O1 | O2 | O3 | Avg. | O0 | O1 | O2 | O3 | Avg. |
| LLM4Decompile-1.3B | 15.30 | 8.26 | 9.36 | 8.38 | 10.33 | 21.79 | 15.23 | 16.17 | 13.70 | 16.72 |
| Nova$_{-CL-HA}$ | 20.73 | 16.16 | 15.03 | 11.19 | 15.78 | 33.55 | 28.12 | 26.96 | 21.01 | 27.41 |
| Nova$_{-HA}$ | 30.58 | 19.88 | 20.58 | 16.40 | 21.86 | 44.75 | 33.13 | 33.31 | 29.82 | 35.25 |
| **Nova** | **37.53** | **21.71** | **22.68** | **18.75** | **25.17** | **49.38** | **34.84** | **36.95** | **32.03** | **38.30** |

Table 6: Recall@1 on BCSD with $K = 50$

| Benchmarks | jTrans | DiEmph | CodeArt | Nova-1.3B | Nova-6.7B |
|---|---|---|---|---|---|
| Binutils | 0.68 | 0.80 | 0.84 | 0.87 | 0.89 |
| Curl | 0.72 | 0.84 | 0.86 | 0.89 | 0.94 |
| ImageMagick | 0.53 | 0.71 | 0.78 | 0.86 | 0.90 |
| SQLite | 0.73 | 0.79 | 0.78 | 0.77 | 0.78 |
| OpenSSL | 0.70 | 0.83 | 0.88 | 0.90 | 0.92 |
| Putty | 0.63 | 0.72 | 0.69 | 0.72 | 0.71 |
| Avg. | 0.67 | 0.78 | 0.81 | **0.84** | **0.86** |

Table 7: Recall@1 on BCSD with $K = 100$

| Benchmarks | jTrans | DiEmph | CodeArt | Nova-1.3B | Nova-6.7B |
|---|---|---|---|---|---|
| Binutils | 0.60 | 0.63 | 0.81 | 0.79 | 0.79 |
| Curl | 0.63 | 0.80 | 0.82 | 0.86 | 0.88 |
| ImageMagick | 0.54 | 0.71 | 0.76 | 0.79 | 0.81 |
| SQLite | 0.62 | 0.72 | 0.74 | 0.73 | 0.72 |
| OpenSSL | 0.60 | 0.80 | 0.87 | 0.88 | 0.90 |
| Putty | 0.58 | 0.64 | 0.64 | 0.65 | 0.64 |
| Avg. | 0.60 | 0.72 | 0.77 | **0.78** | **0.79** |

Table 8: Recall@1 on BCSD with $K = 200$

| Benchmarks | jTrans | DiEmph | CodeArt | Nova-1.3B | Nova-6.7B |
|---|---|---|---|---|---|
| Binutils | 0.51 | 0.64 | 0.74 | 0.73 | 0.73 |
| Curl | 0.57 | 0.77 | 0.78 | 0.83 | 0.84 |
| ImageMagick | 0.39 | 0.51 | 0.67 | 0.73 | 0.75 |
| SQLite | 0.56 | 0.65 | 0.68 | 0.68 | 0.69 |
| OpenSSL | 0.54 | 0.71 | 0.82 | 0.84 | 0.88 |
| Putty | 0.49 | 0.58 | 0.55 | 0.55 | 0.58 |
| Avg. | 0.51 | 0.64 | 0.71 | **0.73** | **0.75** |

Table 9: Recall@1 on BCSD with $K = 500$

| Benchmarks | jTrans | DiEmph | CodeArt | Nova-1.3B | Nova-6.7B |
|---|---|---|---|---|---|
| Binutils | 0.40 | 0.57 | 0.70 | 0.65 | 0.67 |
| Curl | 0.43 | 0.62 | 0.69 | 0.73 | 0.76 |
| ImageMagick | 0.25 | 0.42 | 0.58 | 0.61 | 0.65 |
| SQLite | 0.43 | 0.59 | 0.62 | 0.59 | 0.62 |
| OpenSSL | 0.43 | 0.61 | 0.76 | 0.78 | 0.82 |
| Putty | 0.38 | 0.50 | 0.49 | 0.47 | 0.51 |
| Avg. | 0.39 | 0.55 | 0.64 | **0.64** | **0.67** |

(Figure 5 (c)) embeddings group the assemblies by functionalities more precisely than Nova$_{-HA}$, suggesting that hierarchical attention enhances the training of contrastive learning objectives to learn more effective encoding. Visualization using a different approach, PCA, is shown in Appendix A.7. Analytic experiments on Nova's hierarchical attention is shown in Appendix A.8.

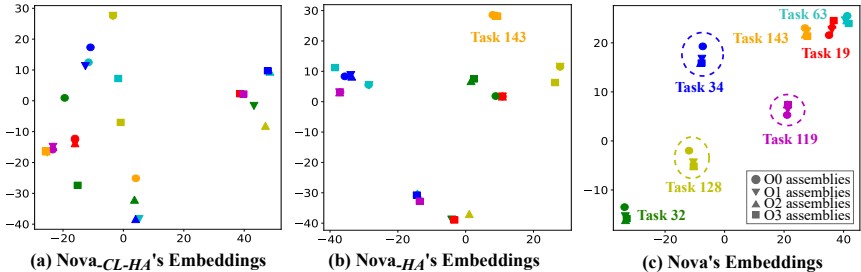

Figure 5: t-SNE analysis of embeddings calculated by Nova$_{-CL-HA}$, Nova$_{-HA}$, and Nova.

# 6 CONCLUSION

This work presents Nova, a generative foundation LLM for assembly code that introduces two key innovations—hierarchical attention and contrastive learning—to tackle the unique challenges of assembly code understanding. Nova builds upon a source code LLM and undergoes further pre-training on approximately 4.3 million collected assembly functions. We fine-tune and evaluate Nova on two downstream tasks: binary code decompilation and binary code similarity detection. Nova achieves up to 14.84 – 21.58% (absolute percentage point improvement) higher Pass@1 and Pass@10 over existing methods in binary code decompilation and up to a 6.17% higher Recall@1 in binary code similarity detection. Looking forward, we believe that the proposed hierarchical attention and contrastive learning methods hold promise for enhancing foundation models in both source code and natural language domains, which we leave as future work.

## ACKNOWLEDGMENTS

This research was supported in part by NSF 1901242 and 2006688, a CFI fund, and J.P. Morgan AI Faculty Research Awards. Any opinions, findings, and conclusions in this paper are those of the authors only and do not necessarily reflect the views of our sponsors.

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

# A   APPENDIX

## A.1   DATA COLLECTION

This section provides additional details of the data collection. To collect assemblies from The-Stack, we attempt to compile 4 million C programs, of which 138.8K is compiled successfully. We do not collect more due to the computation resource limitations.

For the 757.1K and 138.8K source code that successfully compiled into executables (using gcc) from AnghaBench and The-Stack, we disassemble them using objdump. objdump was not able to successfully disassemble all the executables, resulting in some empty assembly code. Thus, the number of O0 − O1 we obtain from each corpus is different and smaller than the number of source codes as shown in Table 1.

There may be alternative ways of collecting assembly code, e.g., using gcc -S to directly dump the assembly code without producing executables. However, a key difference is that the assembly generated by gcc -S does not undergo the linking step. In practice, binary decompilation and analysis are typically performed on executable or linked assembly code, as it includes linker modifications and reflects the final binary structure. Our data collection aligns better with the practical use of assembly. Nevertheless, Nova's approaches should be generalizable to assembly code obtained with a different approach, and the experiments remain as future work.

Figure 6 shows an example of preprocessing the raw assembly code as described in Section 3.1.

```
0:    endbr64                              endbr64                    [INST-0]
4:    push   %rbp                          push rbp                   [INST-1]
5:    mov    %rsp,%rbp                     mov  rsp , rbp             [INST-2]
8:    mov    %rdi,-0x8(%rbp)               mov  rdi , -8 ( rbp )      [INST-3]
c:    mov    %rsi,-0x10(%rbp)              mov  rsi , -16 ( rbp )     [INST-4]
10:   mov    -0x8(%rbp),%rax               mov  -8 ( rbp ) , rax      [INST-5]
14:   mov    (%rax),%edx                   mov  ( rax ) , edx         [INST-6]
16:   mov    -0x10(%rbp),%rax              mov  -16 ( rbp ) , rax     [INST-7]
1a:   mov    (%rax),%eax                   mov  ( rax ) , eax         [INST-8]
1c:   cmp    %eax,%edx                     cmp  eax , edx             [INST-9]
...                                        ...
```


**(a) Raw Assembly**      **(b) Normalized Assembly**



Figure 6: Example of assembly code preprocessing

## A.2   EXPERIMENTAL SETUP DETAILS

This section provides additional details of training. We pre-train Nova starting from DeepSeek-Coder, using the language modeling objective ($L_{lm}$) for one epoch on the C functions and assembly functions collected from AnghaBench and The-Stack corpora. This step uses a batch size of 128, with the input

Table 10: Comparison with applying hierarchical attention on all attention heads, using 1B models.

| Techniques | Pass@1 | | | | | Pass@10 | | | | |
|---|---|---|---|---|---|---|---|---|---|---|
| | O0 | O1 | O2 | O3 | Avg. | O0 | O1 | O2 | O3 | Avg. |
| Nova$_{-HA}$ | 30.58 | 19.88 | 20.58 | 16.40 | 21.86 | 44.75 | 33.13 | 33.31 | 29.82 | 35.25 |
| Nova (hierarchical on all heads) | 32.38 | 18.87 | 20.56 | 16.34 | 22.04 | 45.95 | 32.19 | 32.78 | 29.01 | 34.98 |
| **Nova** | **37.53** | **21.71** | **22.68** | **18.75** | **25.17** | **49.38** | **34.84** | **36.95** | **32.03** | **38.30** |

truncated by a 1,024 tokens limit. The model weights are updated using the AdamW optimizer. The learning rate is $5e^{-5}$, using 1000 steps of warm-up and a cosine decay to adjust the learning rate.

Then, the model is further pre-trained with the combination of language modeling and contrastive learning objectives ($L = L_{lm} + \lambda(L_{fcl} + L_{ocl})$), with $\lambda$ set to 0.1. To train with the functionality contrastive learning objective, we discard any source code that misses any one of O0 − O3 assemblies and also discard the source code whose O2 assembly is the same as its O3 assembly. As a result, this step is only trained for 0.36M data samples for one epoch. The batch size is 64, with the input truncated by a 1,024 tokens limit. The learning rate is $2e^{-5}$ using the AdamW optimizer.

The fine-tuning of both BCD and BCSD uses a batch size of 64, with the input truncated by a 2,048 token limit. Similarly, the learning rate is $2e^{-5}$ using the AdamW optimizer, and the model is fine-tuned for one epoch. During the training using the contrastive learning objectives, and the fine-tuning of BCSD, we use the average of [INST] tokens' last layer hidden states to represent the embedding of a binary function.

For the evaluation of BCSD, we reuse the framework provided by CodeArt Su et al. (2024) to evaluate the binary code similarity detection results once Nova produces the embeddings for functions in the test dataset. For each one of the $K$ functions, the O0 assembly function is used as the query to calculate the cosine similarity between its embedding and the embeddings of all the $K$ O3 assembly functions. Note that Nova does not normalize the embeddings of assembly functions during training and uses $l_2$ distance to calculate the $L_{BCSD}$, which optimize Nova to embed the O0 and O3 assembly functions from the same source code have the smallest $l_2$ distances. When using CodeArt's framework for evaluation which ranks the similarity using the cosine similarity, we normalize Nova's embedding for each assembly function since $l_2$ distance after normalization keeps the same order as cosine similarity (smaller $l_2$ distance means higher cosine similarity).

**Infrastructure** The experiments are conducted on eight NVIDIA RTX A100 GPUs, each with 40GB memory. Our implementation is based on Huggingface's implementation of DeepSeek-Coder [1], PyTorch [2], and DeepSpeed [3].

### A.3 APPLYING HIERARCHICAL ATTENTION ON HALF ATTENTION HEADS

The hierarchical attention mask is always applied on half of the attention heads at each layer in Nova. This ensures the LLM balances the hierarchical knowledge of assembly code and pre-trained knowledge learned by full self-attention.

We conducted experiments applying hierarchical attention to all the attention heads. Results in Table 10 show that when applying hierarchical attention to all the attention heads of transformer layers, the performance does not improve enough and even drops under some settings. This implies that the standard full self-attention mechanism indeed learns knowledge that may not be captured by hierarchical attention. Thus, we only apply hierarchical attention to half of the attention heads in each transformer layer of Nova to balance the knowledge learned by standard and hierarchical attention.

### A.4 ADDITIONAL ABLATION STUDY ON BINARY CODE DECOMPILATION

We provide additional ablation studies on studying the impact of each individual contrastive learning objective. We study two more models:

- Nova$_{-FCL-HA}$: Removing functional contrastive learning and hierarchical self-attention.

---

[1] https://huggingface.co/deepseek-ai/deepseek-coder-1.3b-base
[2] https://pytorch.org/get-started/locally/
[3] https://github.com/microsoft/DeepSpeed

Table 11: Ablation study of Nova-1.3B on HumanEval-Decompile.

| Techniques | Pass@1 | | | | | Pass@10 | | | | |
|---|---|---|---|---|---|---|---|---|---|---|
| | O0 | O1 | O2 | O3 | Avg. | O0 | O1 | O2 | O3 | Avg. |
| Nova$_{-CL-HA}$ | 20.73 | 16.16 | 15.03 | 11.19 | 15.78 | 33.55 | 28.12 | 26.96 | 21.01 | 27.41 |
| Nova$_{-FCL-HA}$ | 22.38 | 16.20 | 16.37 | 13.25 | 17.05 | 36.13 | 29.48 | 30.02 | 23.76 | 29.85 |
| Nova$_{-OCL-HA}$ | 28.44 | 18.87 | 18.53 | 15.76 | 20.40 | 40.28 | 32.33 | 31.80 | 27.05 | 32.87 |
| Nova$_{-HA}$ | 30.58 | 19.88 | 20.58 | 16.40 | 21.86 | 44.75 | 33.13 | 33.31 | 29.82 | 35.25 |
| **Nova** | **37.53** | **21.71** | **22.68** | **18.75** | **25.17** | **49.38** | **34.84** | **36.95** | **32.03** | **38.30** |

- Nova$_{-OCL-HA}$: Removing optimization contrastive learning and hierarchical self-attention.

Table 11 shows that each component of the contrastive learning brings certain improvements to the Pass@1 and Pass@10 on HumanEval-Decompile, and we find the impact of function contrastive learning (FCL) is larger than the impact of optimization contrastive learning (OCL), suggesting that aligning the model's embeddings for assembly code with the same functionality is more useful.

```
<func0>:
0:    endbr64
4:    test   %esi, %esi
6:    jle    48 <func0+0x48>
8:    lea    -0x1(%rsi), %ecx
b:    add    $0x1, %rcx
f:    mov    $0x1, %edx
14:   movss  0x0(%rip), %xmm3
1b:   00
1c:   cmp    %rcx, %rdx
1f:   je     4e <func0+0x4e>
21:   movss  -0x4(%rdi, %rdx, 4), %xmm2
27:   mov    %rdx,%rax
2a:   movaps %xmm2, %xmm1
2d:   subss  (%rdi, %rax, 4), %xmm1
32:   andps  %xmm3, %xmm1
35:   comiss %xmm1, %xmm0
38:   ja     54 <func0+0x54>
3a:   add    $0x1, %rax
3e:   cmp    %eax, %esi
40:   jg     2a <func0+0x2a>
42:   add    $0x1, %rdx
46:   jmp    1c <func0+0x1c>
48:   mov    $0x0, %eax
4d:   retq
4e:   mov    $0x0, %eax
53:   retq
54:   mov    $0x1, %eax
59:   retq
```
**O1 Assembly Code**

```
int func0(float *arr, int n) {
  for (int i = 1; i < n; i++) {
    for (int j = 0; j < i; j++) {
      if (fabs(arr[i] - arr[j]) < 1.0f)
        return 1;
    }
  }
  return 0;
}
```
**GPT-4's Incorrect Decompilation**

```
int func0(const float *x, int n) {
  int i;
  float diff = 1.0f;
  for (i = 1; i < n; i++) {
    if (fabs(x[i] - x[i-1]) > diff)
      return 0;
    return 1;
}
```
**LLM4Decompile's Incorrect Decompilation**

```
int func0(float *a, int n, float e) {
  int i, j;
  for (i = 0; i < n; i++)
    for (j = i + 1; j < n; j++)
      if (fabs(a[i] - a[j]) < e)
        return 1;
  return 0;
}
```
**Nova's Correct Decompilation**

Figure 7: Nova-1.3B correctly decompiles HumanEval-Decompile task 0.

## A.5 BINARY CODE DECOMPILATION CASE STUDIES

Figure 7 shows an example from HumanEval-Decompile (`task_id` 0). Given the `O1`-optimized assembly code, GPT-4 fails to figure out the number of function arguments correctly, missing one important argument "`float e`", and thus produces wrong functionality in the decompiled code. LLM4Decompile-1.3B makes similar mistakes and also misses the inner nested `for` loop. Nova-1.3B correctly decompiles the assembly into source code, where the ground truth is checking if any two elements in the given list `*a` (with size `n`) are close to each other than a given threshold `e`.

Figure 8 shows another more complex example, HumanEval-Decompile `task_id` 20. Nova-1.3B correctly decompiles the source code, successfully figuring that the function is trying to find the two elements that are closest to each other in the given array `*a`, with the result stored in `minmax`.

## A.6 BINARY CODE SIMILARITY DETECTION ABLATION STUDY

Table 12, 13, 14, 15 show the detailed ablation study results of BCSD. Nova wins on the most benchmarks when $K = 100$ or $500$, and ties with Nova$_{-HA}$ when $K = 50$, or $200$.

```
<func0>:                                    void func0(float *a, int n, float *b) {
0:      endbr64                               int i, j;
4:      movss    (%rdi), %xmm0                float min, max;
8:      movss    %xmm0, (%rdx)
c:      movss    0x4(%rdi), %xmm1             min = max = a[0];
11:     movss    %xmm1, 0x4(%rdx)             for (i = 1; i < n; i++) {
16:     test     %esi, %esi                     if (a[i] < min)
18:     jle      8b <func0+0x8b>                  min = a[i];
1a:     lea      -0x1(%rsi), %r9d               else if (a[i] > max)
1e:     mov      $0x1, %r8d                       max = a[i];
24:     mov      %rdi, %rcx                  }
27:     movss    0x0(%rip), %xmm2            b[0] = min;
2e:     00                                  b[1] = max;     LLM4Decompile's Incorrect
2f:     add      $0x1, %r9                  }                       Decompilation
33:     movss    0x0(%rip), %xmm3
3a:     00                                  void func0(float *a, int n, float
3b:     cmp      %r8, %r9                   *minmax) {
3e:     je       82 <func0+0x82>             minmax[0] = a[0];
40:     mov      %r8, %rax                   minmax[1] = a[1];
43:     nopl     0x0(%rax, %rax, 1)
48:     movss    (%rcx), %xmm1               for (int i = 0; i < n; i++) {
4c:     movaps   %xmm1, %xmm0                 for (int j = i + 1; j < n; j++) {
4f:     subss    (%rdi, %rax, 4), %xmm0         float diff = fabs(a[i] - a[j]);
54:     andps    %xmm3, %xmm0                   if (diff < minmax[1] - minmax[0])
57:     comiss   %xmm0, %xmm2                   {
5a:     jbe      6d <func0+0x6d>                  minmax[0] = a[i];
5c:     movss    %xmm1, (%rdx)                    minmax[1] = a[j];
60:     movss    (%rdi, %rax, 4), %xmm1         }
65:     movaps   %xmm0, %xmm2                 }
68:     movss    %xmm1, 0x4(%rdx)           }
6d:     add      $0x1, %rax
71:     cmp      %eax, %esi                 if (minmax[0] > minmax[1]) {
73:     jg       48 <func0+0x48>             float temp = minmax[0];
75:     add      $0x1, %r8                   minmax[0] = minmax[1];
79:     add      $0x4, %rcx                  minmax[1] = temp;
...                                         }                    Nova's Correct
99:     retq                               }                       Decompilation
            O2 Assembly Code
```

Figure 8: Nova-1.3B correctly decompiles HumanEval-Decompile task 20.

Table 12: Ablation study with $K = 50$.

| Benchmarks | Nova$_{-CL-HA}$ | Nova$_{-HA}$ | Nova-1.3B |
|---|---|---|---|
| Binutils | 0.86 | 0.88 | 0.87 |
| Curl | 0.84 | 0.87 | 0.89 |
| ImageMagick | 0.79 | 0.80 | 0.86 |
| SQLite | 0.80 | 0.83 | 0.77 |
| OpenSSL | 0.90 | 0.92 | 0.90 |
| Putty | 0.68 | 0.66 | 0.72 |
| Avg. | 0.81 | 0.83 | 0.84 |

Table 13: Ablation study with $K = 100$.

| Benchmarks | Nova$_{-CL-HA}$ | Nova$_{-HA}$ | Nova-1.3B |
|---|---|---|---|
| Binutils | 0.80 | 0.82 | 0.79 |
| Curl | 0.84 | 0.84 | 0.86 |
| ImageMagick | 0.70 | 0.72 | 0.79 |
| SQLite | 0.74 | 0.78 | 0.73 |
| OpenSSL | 0.89 | 0.89 | 0.88 |
| Putty | 0.59 | 0.60 | 0.65 |
| Avg. | 0.76 | 0.78 | 0.78 |

Table 14: Ablation study with $K = 200$.

| Benchmarks | Nova$_{-CL-HA}$ | Nova$_{-HA}$ | Nova-1.3B |
|---|---|---|---|
| Binutils | 0.71 | 0.74 | 0.73 |
| Curl | 0.80 | 0.73 | 0.83 |
| ImageMagick | 0.61 | 0.63 | 0.73 |
| SQLite | 0.68 | 0.71 | 0.68 |
| OpenSSL | 0.85 | 0.87 | 0.84 |
| Putty | 0.53 | 0.53 | 0.55 |
| Avg. | 0.70 | 0.70 | 0.73 |

Table 15: Ablation study with $K = 500$.

| Benchmarks | Nova$_{-CL-HA}$ | Nova$_{-HA}$ | Nova-1.3B |
|---|---|---|---|
| Binutils | 0.62 | 0.65 | 0.65 |
| Curl | 0.67 | 0.71 | 0.73 |
| ImageMagick | 0.46 | 0.51 | 0.61 |
| SQLite | 0.61 | 0.62 | 0.59 |
| OpenSSL | 0.77 | 0.79 | 0.78 |
| Putty | 0.46 | 0.46 | 0.47 |
| Avg. | 0.60 | 0.62 | 0.64 |

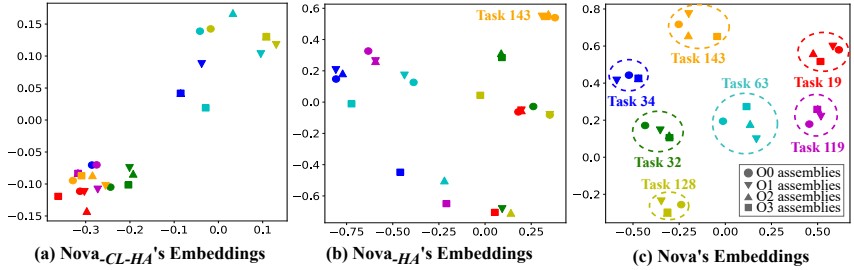

Figure 9: PCA of embeddings calculated by Nova$_{-CL-HA}$, Nova$_{-HA}$, and Nova.

## A.7 ADDITIONAL ANALYSIS OF EMBEDDING

Figure 9 shows the results of PCA of embeddings provided by Nova$_{-CL-HA}$, Nova$_{-HA}$, and Nova, on randomly sampled seven examples, where Nova's embeddings are consistently more distinguishable by functionalities.

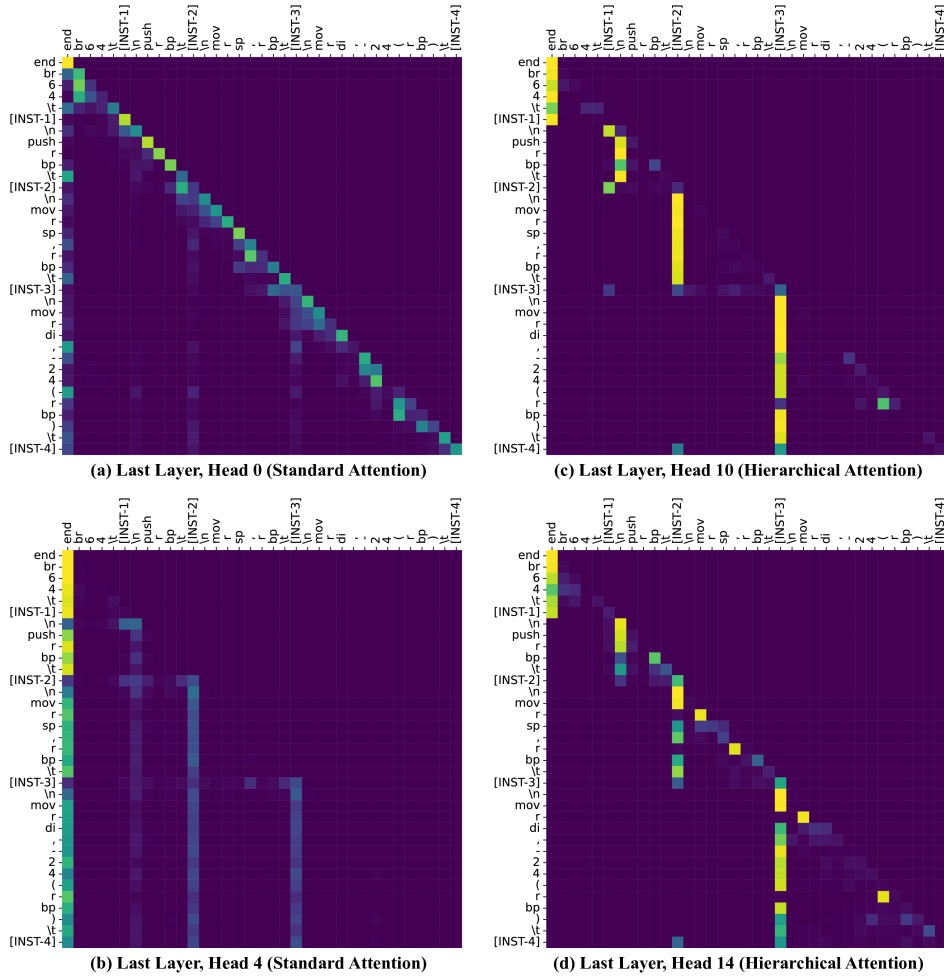

Figure 10: Comparison of attention distribution among standard and hierarchical heads.

## A.8 ADDITIONAL ANALYSIS OF ATTENTION

Figure 10 shows the visualizations of attention weights in the final transformer layer of two select heads with standard attention and two heads with learned hierarchical attention. Standard attention

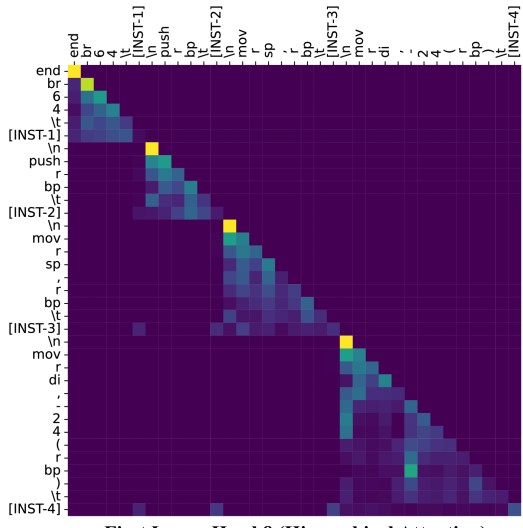

First Layer, Head 8 (Hierarchical Attention)

Figure 11: Learned per-instruction soft attention observed in the lower layers

exhibits two typical patterns, namely diagonal attention (i.e. tokens attending to themselves or nearby tokens, shown in Figure 10 (a)), and broad attention (i.e. a single token attending broadly to the entire sequence, shown in Figure 10 (b)). In contrast, in Nova's hierarchical attention, attention weights are allocated among distinct segments, each corresponding to an instruction (shown in Figure 10 (c)), that focus on tokens comprising that instruction (e.g. opcodes and operands, shown in Figure 10 (d), attentions are paid to "push", "mov", etc.).

Quantitatively, we have determined broad attention accounts for as much as 30% of all attention in standard heads, especially in layers 1-8 (consistent with the findings of (Clark et al., 2019)), whereas in Nova's hierarchical attention, no more than 5% or all attention is allocated to each instruction segment. This validates our goal of learning instruction-aware hierarchical attention in Nova.

In addition, in lower layers, we have observed attention weights to be softly distributed among tokens comprising each instruction (Figure 11), which suggests Nova initially models cross-relations among operation codes and operands in the first few layers, and later pools their summary representation into the [INST] token in the later layers.

## A.9 LIMITATIONS

One limitation is that Nova is X86-specific, as we only collect X86 assembly corpus for pre-training. This design choice is mainly affected by two facts: (1) X86 assembly is used and explored in a wide range of binary tasks (Wang et al., 2022; Su et al., 2024; Xu et al., 2023a; Chen et al., 2022c) compared to other assembly languages, and (2) computation limitations. However, the proposed techniques are independent of X86 assembly. Low information density and compiler optimization are the common challenges of most assembly languages such as X86, ARM, and MIPS. The proposed techniques can be applied to the future development of multi-lingual assembly LLMs. Another potential limitation is the scale of models. We develop Nova-1.3B and Nova-6.7B and demonstrate their advancement in assembly code decompilation and encoding. Developing larger Nova models is promising, but remains as future work due to limited computing resources.

