# OpenReview forum: "Nova: Generative Language Models for Assembly Code with Hierarchical Attention and Contrastive Learning"
_ICLR.cc/2025/Conference — ICLR 2025 Poster_

### Official Review · Reviewer_cjoF · 2024-10-27

**Soundness:** 2
**Presentation:** 2
**Contribution:** 3
**Rating:** 6
**Confidence:** 3

**Summary:**

This paper presents Nova, a generative LLM for assembly code. To effectively learn from assembly with low information density, it uses a novel hierarchical attention mechanism which combines intra-instruction attention, preceding-instruction attention and inter-instruction attention. It further utilizes contrastive learning to better learn the semantics of assembly code from different optimization levels. The authors demonstrate the superiority of Nova over the baselines on binary code decompilation and code similarity detection tasks.

**Strengths:**

1. LLMs for binary code is an important topic to study
2. This work proposes new methods to train Nova based on the properties of assembly code, which is clearly motivated.
3. The proposed models show a clear improvement in binary code decompilation.

**Weaknesses:**

1. The comparison on code similarity detection may not be fair. For example, CodeArt uses 12 transformer blocks with 768 hidden dimensions, whose size is smaller than Nova-1B. The authors should compare Nova with the baseline under a similar size with the same pre-training data to demonstrate the superiority of Nova on code similarity detection. For the current result, we can find that compared with CodeArt, Nova actually does not show a significant improvement (e.g. both are 0.64 for Nova-1B under K=500). So it is in question whether Nova is indeed better for code similarity detection.
2. The experiments for Comparison with Techniques Handling Long Input are confusing. Specifically, it has the following problems:

a) What is "Nova’s Fine-Tuning" in Table 3? It seems Nova does not have something special in terms of fine-tuning. Does it just mean fine-tuning with hierarchical attention or also with Nova's pretraining as suggested in Line 360?

b) What is the average token length for downstream tasks before truncation? The authors want to claim Nova is better at solving long input challenges. But I see from the Appendix that Nova uses the input length as 1024 tokens during pre-training and 2048 for fine-tuning. It may be hard to claim this length to be "long-context". Considering that assembly code should be much longer than source code and Granite-3B-Code-128K can handle 128K input tokens at most, have you tested in the benchmarks where the input context is longer, e.g. 8k/32k/128k?

3. The presentation of the paper can be improved. Specifically, a) Line 281 is unclear. The authors should clearly state that their pre-training contains two stages and the loss in Line 240 is used in the second stage. b) The ablation study should be separated into new subsections instead of mixing with Section 4.1 c) The equations are not numbered.

**Questions:**

1. See weakness 1,2
2. Could you provide more details about how to construct $F$ in practice used in Functional CL?
3. The authors state that hierarchical attention is only applied to half of the attention head. Since different attention heads can learn different features, I wonder if this setup is robust to the selection of the attention heads?
4. Would the pre-trained models (Nova-1B, 6B) be public available?

---

> ### Author Response · Authors · 2024-11-22
>
> We thank the reviewer for the insightful review and questions.
>
> ### 1. Improvement on BCSD
>
> We acknowledge that Nova’s improvement on BCSD is not as significant as on BCD, especially given that Nova has a larger model size.
>
> Yet, binary code similarity is a much better-explored domain than binary code decompilation. There has been more related work from software engineering, and security domain design models for it. The baselines we compare (jTrans, DiEmph, and CodeArt) have already achieved impressive performance. By contrast, end-to-end binary code decompilation is a very new task and LLM4Decompile is the only existing SOTA baseline we can find. Thus, we think there is less space to improve on binary code similarity than binary code decompilation, and that Nova outperforming existing SOTAs is valuable and hard.
>
> Besides, those specialized BCSD models we compare with are encoder transformers. The bidirectional attention benefits their performance on the BCSD task as it is essentially an encoding task. Using decoder-only LLM for encoding tasks such as the BCSD task is less common and harder. Our ablation study (in Appendix Table 12-15) also shows that without Nova’s designs, simply training DeepSeek-Coder-1B cannot outperform existing SOTAs on BCSD.
>
> Then why do we explore Nova (a decoder-only LLM) on BCSD? Given that decoder-only LLM is getting much more popular and promising, and the latest advanced LLMs are almost all decoder-only (Llama, StarCoder, etc), Nova shows the potential of building on top of the latest decoder-only LLMs for encoding tasks such as BCSD so that we can make good use of those pre-trained decoder-only LLMs.
>
>
> ### 2. Table 3
>
> In Table 3, Nova’s fine-tuning means applying hierarchical attention and using contrastive learning objectives. We want to show that, our approach brings improvement over standard fine-tuning on multiple backbones (both DeepSeek-Coder 1.3B and 6,7B, as well as Granite-3B-Code).
>
> Maybe it is clearer to call it “Granite + Nova’s Approaches”.
>
> ### 3. Token Length
>
> In the HumanEval-Decompile benchmark, the longest assembly function has 3240 tokens, and the average is 698 tokens. We do not truncate during inference and we truncate during training mainly for speeding up and saving GPU memory.
>
> Our point is not purely for longer input, since the challenge of assembly code is longer input + low information density. The model has to summarize the tokens at a higher level to extract functional semantics.
>
>
> ### 4. Details of Functional CL
>
> The F in the Functional CL (L 264) is a set of functions in the training data. For example, in practice, we have a batch size of 64 during the training with Functional CL, then F is a set of 64 unique functions, and each function has a source code, O0, O1, O2, and O3 assembly format.
>
> Then we can calculate the loss $L_{fcl}$ (L 274) among this batch and perform backward propagation.
>
>
> ### 5. Apply of Hierarchical Attention
>
> In Appendix Table 10, we show that without hierarchical attention or applying hierarchical attention on all the attention heads are both not good. That’s why we choose to apply on half of the attention heads.
>
> Although a more fine-grained study could be conducted (e.g., applying on a quarter of heads and so on), given that each training of Nova requires nearly a thousand GPU hours, we are unable to thoroughly search the design choices.
>
> Intuitively, this is a trade-off between the effectiveness of hierarchical attention and the pre-training knowledge that is kept in the original attention heads.
>
> ### 6. Availability
>
> Yes, we plan to release the model weights, and code (for hierarchical attention implementation) after the notification.

---

> > ### Comment · Reviewer_cjoF · 2024-11-26
> >
> > Thanks for the authors' response. I think it basically clarifies my questions.

---

### Official Review · Reviewer_v7d8 · 2024-11-03

**Soundness:** 3
**Presentation:** 3
**Contribution:** 2
**Rating:** 5
**Confidence:** 3

**Summary:**

This paper introduces a generative model, Nova, tailored for assembly code tasks. Nova employs a hierarchical attention mechanism and is trained using contrastive learning objectives. This paper evaluates its effectiveness on two assembly code tasks.

**Strengths:**

Strengths:
+ The topic is interesting and important, addressing large language model (LLM) comprehension of assembly code.
+ The paper is well-structured and easy to follow.

**Weaknesses:**

Weaknesses:
- Comparison may be unfair due to different fine-tuning practices.
- Evaluation of individual components is insufficient.
- Generalization assessment is lacking.

(1) Unfair Comparison: Nova is evaluated on two tasks, with fine-tuning applied specifically for each. However, the baseline models (such as Table 2) do not undergo the same fine-tuning for the tasks, leading to a potentially unfair comparison.

(2) Component Evaluation: Nova’s hierarchical self-attention mechanism consists of three components, yet the paper lacks detailed performance assessments for each part. Despite a reasonable design, their individual impact remains unexamined.

(3) Contrastive Learning Objectives: The contrastive learning objectives contain two distinct components. Further evidence is necessary to substantiate the utility of each objective. Additionally, the contrastive learning approach depends on the available optimization levels. Handling unseen optimization levels at inference should be discussed.

(4) Normalization Process: In the data collection section, a normalization step is applied, but its relevance or benefit to Nova’s training is unclear.

(5) Results across different optimization levels should be explored—e.g., training on O0, O1, O2 and testing on O3.

(6) Random Sampling in BCSD Task: The BCSD task employs random sampling, yet statistical results are missing. Reporting such results would reduce the impact of randomness on performance claims.

**Questions:**

Please check my concerns in the weakness section.

---

> ### Author Response · Authors · 2024-11-22
>
> We thank the reviewer for the insightful review and questions.
>
> ### 1. Fairness of Comparison
>
> In our comparison with other models on decompilation, except for LLM4Decompile and Nova (these two are fine-tuned for this task), the other models are prompted with three examples as few-shot learning. Especially for GPT models, few-shot learning is the common practice of applying them to downstream tasks. We show that Nova-1B outperforms the most powerful models (as these are the most powerful commercial or open-source LLMs at the time) by a significant margin, highlighting the contribution of building such a foundation model for assembly code.
>
> Besides, **our main focus is the comparison with LLM4Decompile**. LLM4Decompile  is a closely related baseline that also fine-tuned the base backbone (DeepSeek-Coder) using assembly code collected from AnghaBench. Nova-1B significantly outperforms LLM4Decompile showing that it is indeed Nova’s design (hierarchical attention + contrastive learning) that brings the improvement.
>
> ### 2. Contribution of Each Component of Hierarchical Attention
>
> We are only able to design the hierarchical attention empirically, as the build of Nova models is expensive, requiring thousands of GPU hours.
>
> Yet, we compare Nova’s hierarchical attention design with LongCoder’s attention, showing that Nova’s attention design works better on assembly code.
>
> ### 3. Contribution of Each CL Objective
>
> We conduct additional ablation studies to study the impact of each CL objective (FCL for functional contrastive learning, OCL for optimization contrastive learning). Below is the result:
>
> Nova$_{-CL-HA}$ means no CL and hierarchical attention (HA), basically standard fine-tuning.
>
> Nova$_{-FCL-HA}$ means adding OCL.
>
> Nova$_{-OCL-HA}$ means adding FCL.
>
> Nova$_{-HA}$ means adding both FCL and OCL.
>
> | Model | O0 | O1 | O2 | O3 | Avg |
> |-|-|-|-|-|-|
> | Nova$_{-CL-HA}$ | 20.73 | 16.16 | 15.03 | 11.19 | 15.78 |
> | Nova$_{-FCL-HA}$  | 22.38 | 16.20 | 16.37 | 13.25 | 17.05 |
> | Nova$_{-OCL-HA}$ | 28.44| 18.87 | 18.53 | 15.76 | 20.40 |
> | Nova$_{-HA}$ | 30.58 | 19.88 | 20.58 | 16.40 | 21.86 |
> | Nova-1B | 37.53 | 21.71 | 22.68 | 18.75 | 25.17 |
>
> From this result, we can see that both FCL and OCL bring improvement. Yet, it is clear that FCL indeed has a bigger impact than OCL. We have added this result to the Appendix in our updated version of the PDF.
>
>
> ### 4. Normalization
>
> The design of the normalization process is based on the following insights and requirements:
>
> * Existing code LLMs’ tokenizers do not fit the assembly code very well. For instance: `’(%rax),%rbx’` is tokenized to `‘(‘ ‘%’ ‘ra’ ‘x’ ‘),’ ‘%’ ‘rb’ ‘x’`, where `’),’` is considered as one token. This does not properly show the semantics, since in X86 Assembly, parentheses are explicitly used to denote memory addressing while commas are used to separate operands. As we cannot re-train the tokenizer, we normalize the assembly code to make the tokenization better reflect assembly code semantics. We add white space to separate `‘(‘`, `‘)’` and `‘,’`, so that they are always considered as separate tokens.
>
> * We remove all the hexadecimal values as LLMs are very poor at understanding hexadecimal values.
>
> * Instead, we use the `[INST-i]` tokens to index the instructions. In decoder-only generative LLM, we want them to summarize the semantics of each instruction and thus have to put them at the end of each instruction since decoder-only LLM only has single-direction attention.
>
> ### 5. Train on O0, O1, O2 and Test on O3
>
> We acknowledge this could be a more challenging setting and interesting future work. Yet, most existing baselines (LLM4Decompile for decompilation, jTrans, CodeArt for similarity detection) we compare with use the same setting as us, so we think this research question is out of the scope of this paper.
>
> ### 6. Randomness of Sampling
>
> The randomness of sampling during generation is a common issue for other tasks such as code generation. We think the metric Pass@K [1] already handles the problem since Pass@K measures the expectation/chance of getting correct decompilation if we let the model generate K times (e.g., Pass@1 means the expectation of having one correct decompilation if the model is generated once, and Pass@10 means the expectation of having one correct decompilation if the model is generated 10 times).
>
> In our evaluation, we let each model sample 20 generations and then calculate Pass@1 (i.e., how many of the 20 generations are correct?) and Pass@10 (i.e., if I pick 10 generations from the 20, what is the chance of having at least one correct decompilation?). We think reporting Pass@K with sampling generation is a common practice and the result is statistical.
>
> [1] https://arxiv.org/abs/2107.03374

---

### Official Review · Reviewer_n2JL · 2024-11-04

**Soundness:** 3
**Presentation:** 4
**Contribution:** 3
**Rating:** 8
**Confidence:** 3

**Summary:**

This paper presents Nova, a generative language model specifically designed for assembly code, addressing unique challenges posed by the low information density and diversity in assembly syntax due to compiler optimizations. Nova introduces a hierarchical attention mechanism and employs contrastive learning to improve the model's understanding of assembly semantics across diverse optimization levels. Trained on a large assembly corpus, Nova outperforms existing techniques in tasks like binary code decompilation and binary code similarity detection, showing  improvements in Pass@1 and Recall@1 rates over state-of-the-art models.

**Strengths:**

1. Clear Writing and Novel Application: The paper is well-written and easy to follow. The idea of applying hierarchical attention to assembly code is interesting and novel. While hierarchical attention is commonly used in NLP tasks, applying this mechanism to assembly code is, to the best of my knowledge, unprecedented.

2. Promising Results: The evaluation results are promising. Nova demonstrates substantial improvements in both decompilation accuracy and similarity detection compared to existing models, validating its approach with strong experimental evidence.

**Weaknesses:**

Generalizability: The model is trained exclusively on x86 assembly code, which may limit its generalizability to other assembly languages, such as ARM or MIPS.

Realism of Evaluation Settings:

(1) The decompilation prompt requires optimization level information, but it is unclear if this information is accessible in stripped binaries.

(2) For baseline models like GPT, fine-tuning with additional data isn’t necessary, raising questions about the fairness of the comparison. If GPT were given a few-shot learning setup or fine-tuned using OpenAI’s API, could it still be outperformed by the proposed approach?


Related Work: The paper omits discussion of several relevant works, which could provide a broader context for its contributions.

[1] Debin: Predicting Debug Information in Stripped Binaries. CCS 2018

[2] {DIRE}: A Neural Approach to Decompiled Identifier Renaming. ASE 2019

[3] Learning to Reverse DNNs from AI Programs Automatically. IJCAI 2022

[4] Asm2Vec: Boosting Static Representation Robustness for Binary Clone Search against Code Obfuscation and Compiler Optimization. S&P 2019

[5] Neural Network-based Graph Embedding for Cross-Platform Binary Code Similarity Detection. CCS 2017.

[6] ecompiling x86 Deep Neural Network Executables. Security 2023.

**Questions:**

For binary similarity detection, compilers may inline functions or eliminate them altogether. How does your approach handle such scenarios?

If additional information (e.g., execution traces) were provided to GPT, or if iterative interaction with GPT were allowed, could the proposed approach still outperform a GPT-based model?

---

> ### Author Response · Authors · 2024-11-22
>
> We thank the reviewer for the insightful review and questions.
>
> ### 1. Optimization Level During Inference
>
> We conduct an evaluation of decompilation where the optimization level is not provided to the model. Fortunately, Nova is able to keep the highest performance without the optimization information.
>
> | Model | O0 | O1 | O2 | O3 | Avg |
> |-|-|-|-|-|-|
> | Nova-1B without optimization info. | 37.23 | 20.98 | 22.42 | 19.05 | 24.92 |
> | Nova-1B | 37.53 | 21.71 | 22.68 | 18.75 | 25.17 |
>
> Without the optimization information, the averaged Pass@1 of Nova-1B drops slightly from 25.17 to 24.92, yet it still significantly outperforms other baselines. We also find that without optimization information, Nova-1B actually performs slightly better on decompiling O3 assembly, which could be the variance of experiments.
>
> We think one possible reason why Nova is able to handle decompilation without optimization information, is that during the pre-training (language modeling, contrastive learning) stage, the assembly code is input without any optimization information, Nova is trained to predict assembly instructions and encoding assembly code without knowing the optimization level. This training enables Nova to perform well on decompilation without knowing the optimization information.
>
> ### 2. GPT Baseline
>
> * We want to clarify that during the inference of binary code decompilation, except for LLM4Decompile and Nova (these two are fine-tuned for the task), we provide the other baseline models with three examples for few-shot learning. Thus, the GPT models have three-shot examples.
>
> * We conduct a one-round self-debug experiment using GPT-4o, letting GPT-4o revise the decompilation if it fails the test cases. The Pass@1 result is shown below:
>
> | Model | O0 | O1 | O2 | O3 | Avg |
> |-|-|-|-|-|-|
> | GPT-4o | 21.34 | 18.29 | 14.48 | 13.05 | 16.79 |
> | GPT-4o + Self-Debug | 25.46 | 21.08 | 16.73 | 14.19 | 19.37 |
> | Nova-1B | 37.53 | 21.71 | 22.68 | 18.75 | 25.17 |
>
> Although GPT-4o is able to revise some incorrect decompilation if we provide the execution feedback to it, the Pass@1 after such revision is still much worse than Nova-1B which is designed and fine-tuned for assembly code.
>
> ### 3. Related Works
>
> We thank the reviewer for pointing out additional related work, and we have added them to the "Binary Models" section in the related work part in our updated version of the PDF.

---

### Official Review · Reviewer_hTSw · 2024-11-04

**Soundness:** 3
**Presentation:** 3
**Contribution:** 2
**Rating:** 3
**Confidence:** 4

**Summary:**

This paper presents Nova, a generative language model specifically crafted for understanding and generating assembly code. Nova integrates hierarchical attention mechanisms with contrastive learning to effectively capture the semantics of code. The hierarchical attention mechanism focuses on intra-instruction, preceding-instruction, and inter-instruction relations, while contrastive learning ensures that functionally equivalent code, even with different optimizations, is similarly represented. The model is evaluated on two key tasks: decompilation (recovering high-level source code from assembly) and binary code similarity detection (BCSD) (measuring the similarity between binary code functions). Nova shows superior performance in both tasks, excelling in decompilation by accurately generating source code from optimized assembly, and achieving high recall in BCSD by effectively identifying similar code across different optimization levels.

**Strengths:**

1. This paper is well-structured and easy to follow. Concepts such as hierarchical attention and contrastive learning are clearly explained.
2. The paper proposes a new method for encoding assembly code by using a Hierarchical Attention Mechanism to effectively capture the semantics of assembly instructions, while employing Contrastive Learning to ensure that functionally equivalent assembly code, even at different optimization levels, is represented similarly. This novel combination allows the model to robustly understand and learn from diverse assembly code structures.
3. The paper conducts a broad range of experiments across multiple tasks and datasets, providing comprehensive evidence of the model’s effectiveness.
4. Despite its specialized focus on assembly code, Nova's hierarchical attention is compatible with standard self-attention mechanisms, allowing it to seamlessly integrate and benefit from advancements in base models and code generation models.

**Weaknesses:**

1. Unclear motivation for introducing several inductive bias by Hierarchical Attention Mechanism. While the added attention mask inductive bias shows promising results in the BCD task, its impact in the BCSD task is minimal. This discrepancy raises questions about why the inductive bias performs well in one task but fails to offer significant improvements in the other.
2. Lack of Design Discussion. The paper lacks sufficient discussion on key design components like Preceding-Instruction Attention and Optimization Contrastive Learning (CL). Without Preceding-Instruction Attention, the attention design is quite similar to CodeART, raising questions about the novelty and contribution of the approach.

**Questions:**

1. The paper argues that preceding-instruction attention helps avoid reuse of the same register (e.g., "eax") immediately after it is used in the previous instruction. However, this motivation is questionable because it does not explain how further subsequent instructions are prevented from reusing the same register. A more straightforward solution could be achieved with inter-instruction attention, as it can attend to all previous instructions, which raises the concern of functional overlap between preceding-instruction attention and inter-instruction attention, thus potentially making the preceding-instruction attention redundant.
2. While Nova-1B and Nova-6B are much larger than CodeART, their performance gains in BCSD are limited. For example, in the k=100 case, CodeART sees a 17% improvement over JTrans with attention regularization, but Nova's improvement is only marginal, from 0.76 to 0.78 (as shown in Table 12). This suggests that adding hierarchical attention and other inductive biases provides limited benefits when scaling the model, and Tables 11-14 show that removing hierarchical attention does not lead to significant performance drops, questioning its overall necessity. And also in Table 5, the improvement brought by contrastive learning is much higher than the Hierarchical Attention.
3. In the paper's analysis of attention distribution (Figure 10), the standard attention frequently converges on the first token, a phenomenon known as attention sink [1]. This behavior is also evident in the analysis of hierarchical attention (Figure 10(c, d)), where each token strongly attends to the first token within its attention mask, specifically the [INST-(x-1)] token, which represents the summary of the previous instruction. But it is not common when human try to interpret the functionality of each individual instruction. Furthermore, the justification for the Hierarchical Attention Mechanism —which selectively uses specific attention heads to represent the best attention maps—is somewhat ad hoc and lacks a clearer rationale.
4. The Hierarchical Attention Mechanism introduced in this paper represents a strong inductive bias; however, the underlying insights behind this inductive bias are not clearly explained. Additionally, the mechanism bears a striking resemblance to the Attention Regularization used in CodeART, with the primary difference being the absence of Preceding-Instruction Attention in CodeART. The effectiveness of this additional attention component has also been called into question earlier in the reivew, casting some doubt on its true contribution to the overall performance.
5. While the use of contrastive learning aligns well with the BCSD task—improving performance by ensuring that functionally similar binaries, even across different optimizations, are represented similarly—it's less clear how this objective enhances the model's ability in decompliation. The training goal focuses on increasing the similarity of tokens from the same function but compiled with different optimization settings. However, this doesn't seem directly aligned with the ultimate goal of recovering executable source code, which requires more precise structural and semantic understanding beyond just token similarity across optimization levels. It would be greatly appreciated if the authors could provide some intuition as to why this approach can lead to improvements in decompliation.
6. The authors introduced a novel optimization contrastive learning approach for the BCSD task, which had not been previously applied in the previous works, which commonly use the InfoNCE loss (line 220) or the triplet loss. As it is not discussed with deeper detail in the paper, it raises the question of whether these gains are substantial enough to justify the added complexity and whether this approach could be effectively generalized to improve other models in BCSD tasks.

[1]: Efficient Streaming Language Models with Attention Sinks

---

> ### Author Response · Authors · 2024-11-22
>
> We thank the reviewer for the insightful review and questions.
>
> ### 1. Design Choice of Preceding-Instruction Attention
>
> There are two reasons for using preceding-instruction attention:
> * Intuitively, we separate the preceding-instruction attention from the inter-instruction attention, since inter-instruction attention is more about higher-level semantics. For the example in Figure 1 (a) and (b). The five instructions from 10 to 1c are corresponding to the if-condition in the source code. Such functional semantics are different from the more fine-grained dependencies in the compiler’s view (allocation of registers, maintaining of stack).
> * Another reason is that Nova is designed for decoder-only generative LLM, where the attention is single-directional. The preceding-instruction attention enables every token in an instruction capture the dependencies with previous instructions. While the inter-instruction attention is only among the `[INST-i]` tokens (due to the design that we want this to capture functional semantics across multiple instructions), and thus the other tokens in the instruction do not have access to the inter-instruction attention.
>
> ### 2. Improvement of BCSD
>
> We agree the improvement on the binary code similarity task is less significant than that on the binary code decompilation task. Yet, binary code similarity is a much better-explored domain than binary code decompilation. There has been more related work from software engineering, and security domain design models for it. The baselines we compare (jTrans, DiEmph, and CodeArt) have already achieved impressive performance. By contrast, end-to-end binary code decompilation is a very new task and LLM4Decompile is the only existing SOTA baseline we can find. Thus, we think there is less space to improve on binary code similarity than binary code decompilation, and that Nova outperforming existing SOTAs is valuable and hard.
>
> Besides, those specialized BCSD models we compare with are encoder transformers. The bidirectional attention benefits their performance on the BCSD task as it is essentially an encoding task. Using decoder-only LLM for encoding tasks such as the BCSD task is less common and harder. Our ablation study (in Appendix Table 12-15) also shows that without Nova’s designs, simply training DeepSeek-Coder-1B cannot outperform existing SOTAs on BCSD.
>
> Then why do we explore Nova (a decoder-only LLM) on BCSD? Given that decoder-only LLM is getting much more popular and promising, and the latest advanced LLMs are almost all decoder-only (Llama, StarCoder, etc), Nova shows the potential of building on top of the latest decoder-only LLMs for encoding tasks such as BCSD so that we can make good use of those pre-trained decoder-only LLMs.
>
> ### 3. Difference Between Nova’s Hierarchical Attention and CodeArt’s Attention Regularization
>
> There is one fundamental difference between Nova's and CodeArt’s attentions. CodeArt is designed for Encoder LLMs (e.g., BERT) with bidirectional attention, while Nova is designed for Decoder-only generative LLMs with single-directional attention. This design choice is that decoder-only LLM shows more promise in generation tasks, and the most advanced LLMs are typically decoder-only.
>
> Thus, Nova’s attention has to use the proceeding-instruction attention to let every token in an instruction capture dependencies with previous instructions.
>
> In addition, Nova’s design is compatible with the standard self-attention used for source code (Figure 3 (c)), as we want to build Nova’s approach as a plug-in that can work with any transformer-based, decoder-only LLM to make use of their pre-training knowledge. While CodeArt’s design only works for assembly code and cannot handle input mixes assembly and source code or natural language.
>
> ### 4. How Contrastive Learning Improves Decompilation
>
> Our insight is that contrastive learning enables the model to produce similar embeddings (i.e., the hidden states from the last transformer layer for `[INST-i]` tokens) for assembly code from the same source code. As LLMs understand the O0 assembly code the best (O0 assembly is decompiled with the highest pass@k), such aligning enables the LLMs to link the O1-O3 assembly with the O0 assembly and potentially learn the O1-O3 assembly better.
>
> Besides, during the decompilation, the LLM generates the decompilation given the assembly code as input, i.e., the generation of decompiled source code is conditioned on the hidden states/embeddings of the input assembly code. Thus, we think better embeddings can lead to better decompilation.
>
> ### 5. Generalizability
>
> Nova’s approach is generalizable to different assembly languages such as ARM or MIPS, since they share a similar challenge as X86. Nova’s approach can also be used on any transformer-based, decoder-only generative LLM.
>
> [1] https://aclanthology.org/2023.acl-long.355/

---

> > ### Comment · Reviewer_hTSw · 2024-11-27
> >
> > Thank you for the response. Here are some comments about your response.
> >
> > 1. Design Choice of Preceding-Instruction Attention
> >
> > The authors mention that "inter-instruction attention is more about higher-level semantics." However, as evidenced by Figure 6 in CodeArt, "token level inter-instruction" fine-grained attention is not necessarily required to achieve excellent performance.
> >
> > Additionally, this mechanism could be perceived as a sliding window with window size equals two at the instruction level. Why not extend the sliding window to the basic block level instead?
> >
> > The authors also state that "The preceding-instruction attention enables every token in an instruction to capture dependencies with previous instructions." However, this capability is also present in inter-instruction attention, where [INST-i] can attend to [INST-(i-1)], and through positional embeddings, the model can discern that this token is the preceding one.
> >
> > 2. Comparison of Performance on BCSD
> >
> > The authors claim that "simply training DeepSeek-Coder-1B cannot outperform existing SOTAs on BCSD," which is an unfair comparison. In all attempts to use decoders as embedding models, as shown in [1], it is not common practice to use the base model's results directly for downstream tasks. Instead, with the incorporation of contrastive learning, a widely used embedding training method, $Nova_{−HA}$ achieves significantly high results.
> >
> > From the experiments in the appendix regarding the impact of each contrastive learning objective, it is evident that FCL has an effect, but it is not as significant as OCL, which is a common training method for BCSD. The applicability of FCL should also be considered, as monotonicity may not always be present under other compiler optimizations such as `-Os`, `-O3`, `-Ofast`. These optimization levels are not always directly comparable, affecting the generalizability of this training loss.
> >
> > 3. Distinction Between Nova's Hierarchical Attention and CodeArt's Attention Regularization
> >
> > The authors highlight that "one fundamental difference between Nova's and CodeArt's attentions" is that CodeArt uses bidirectional attention, whereas Nova employs unidirectional attention. This is not the core distinction. CodeArt's paper does not specify that it is limited to bidirectional attention. Current models do not handle the common jmp instruction in assembly code, including Nova. The authors did not address how Preceding-Instruction Attention is managed with jmp instructions. Consider the following assembly example:
> >
> > ```
> >   jmp .label2
> > .label:
> >   mov rax, rbx
> >   cmp rax, 0
> >   je .label
> > .label2:
> >   ret
> > ```
> >
> > In this example, the preceding instruction for `mov rax, rbx` should be `je .label`, but with unidirectional attention, it is impossible to add Preceding-Instruction Attention from `mov rax, rbx` to `je .label`. However, Nova maintains good performance in programs with numerous control flow jumps, indicating that unidirectional attention does not hinder understanding. Therefore, the core difference between CodeArt and Nova is not the attention mechanism's directionality.
> >
> > Additionally, a simple search reveals that mixing unidirectional and bidirectional attention in LLMs is feasible [2][3].
> >
> > 4. Generalizability
> >
> > It is worth noting that the original review did not raise questions about generalizability. However, the authors addressed this in their rebuttal.
> >
> > 5. Overall
> >
> > Considering the aforementioned points, the authors have not clearly demonstrated the innovation of the attention mechanism in the paper. The role of Preceding-Instruction Attention and its distinction from CodeArt are not well-articulated. Nevertheless, given the promising results Nova achieves in improving binary code decompilation, I am open to reconsidering the score.
> >
> > [1] Improving Text Embeddings with Large Language Models (http://arxiv.org/abs/2401.00368)
> >
> > [2] Bitune: Bidirectional Instruction-Tuning (https://arxiv.org/abs/2405.14862v1)
> >
> > [3] Transfusion: Predict the Next Token and Diffuse Images with One Multi-Modal Model (https://arxiv.org/abs/2408.11039)

---

> ### Author Response · Authors · 2024-11-29
>
> Thanks for the follow up. We generally agree with the reviewers, but we want to clarify a few disagreements.
>
> ### **1 & 3: CodeArt’s Limitation on Unidirectional Attention, which Motivates Preceding-Instruction Attention**
>
> CodeArt’s attention is indeed limited for the unidirectional attention (decoder-only) model. The reviewer says: “token level inter-instruction" fine-grained attention is not necessarily based on CodeArt’s Figure 6”, which seems not entirely true given their use of the `<cls>` token.
>
> Considering two instructions, CodeArt puts `<inst>` at the beginning of every instruction and puts a `<cls>` token at the beginning of the entire binary program:
> ```
> <cls>
> <inst> mov ( rdi ) , ecx
> <inst> mov ( rsi ) , edx
> ```
>
> In CodeArt’s design, the `<inst>` tokens have access to all the other `<inst>` tokens, so they are responsible for cross-instruction dependencies, our inter-instruction attention is similar to this.
>
> But, the other tokens such as `mov`, `rsi` do not have access to other instructions, so the embeddings of these tokens inside each instruction may not consider the context. To address this, CodeArt gives every token inside the instruction access to the `<cls>`, and `<cls>` has access to all tokens in the binary program. Thus the `rsi` in the 2nd instruction can capture the dependencies with `rdi` in the 1st instruction, through `rsi -> <cls> -> rdi`, and vice versa. The CodeArt authors call it “global context” (Figure 6b in CodeArt paper), where each token does have dependencies with every other token through this `<cls>` token.
>
> **However, this design is not applicable to the decoder-only models**, because in decoder-only models, there cannot be such a `<cls>` token as a “hub” to transfer attention among tokens in different instructions:
> * if we put <cls> at the beginning, the <cls> token won’t have access to any tokens.
> * if we put <cls> at the end, the other tokens won’t have access to it.
> * if we add <cls> in the middle, the <cls> won’t have access to tokens after it, and the tokens before <cls> won’t have access to <cls>, still lack of circular attention transfer.
>
> CodeArt’s design actually shows we do want tokens inside instructions to have some attention to the context, although the intra-instruction attention within each individual instruction should be dominant. This benefits the hidden states of the non-<inst> tokens.
>
> **Nova achieves this with preceding-instruction attention**.  See the same example binary but in Nova’s format:
>
> ```
> mov ( rdi ) , ecx [inst-1]
> mov ( rsi ) , edx [inst-2]
> ```
>
> The `rsi` could have access to `rdi` through `rsi -> [inst-1] -> rdi`. `rdi` cannot access `rsi` anyway due to the limitation of unidirectional attention.
>
> The reviewer proposes using window attention with a window size equal to two instructions, which indeed could be a promising design. But this is conceptually similar to our preceding-instruction attention together with the intra-instruction attention.
>
> We also agree that other mixed designs could be a choice. We would like to discuss them in the related work sections in the final version. However, we cannot explore these solutions in the scope of this work.
>
> ### **2: A Small Misunderstanding of Table 11**
>
> In Table 11, `Nova−OCL−HA` (the better one) is without OCL so it is actually using FCL. We did “components removal” in the ablation study design. Our conclusion is that FCL has a bigger impact than OCL.
>
> FCL aligns the functionality. Given the same functionality, no matter with `O0-O3, Os, Ofast` optimization, the assembly should all be grouped together. This is a common method for BSCD, and this is generalizable to different optimizations.
>
> OCL considers the optimization monotonicity, which has less impact and is less generalizable. But Table 11 still shows that adding OCL leads  to better results, especially with O0-O3 binaries. So one should not ignore the contribution of OCL during training.
>
> Except for this small misunderstanding, we agree with the reviewer's opinion on the BSCD results.
>
> We appreciate the reviewer’s acknowledgment of Nova’s promising results on the decompilation task.

---

### Official Review · Reviewer_MUPt · 2024-11-06

**Soundness:** 3
**Presentation:** 2
**Contribution:** 3
**Rating:** 6
**Confidence:** 4

**Summary:**

The paper presents a way of training an LLM to improve its performance on tasks that require understanding of assembly code, in particular code decompilation, and assembly code similarity detection.

This is achieved by several contributions:
1. A multi-way, parallel corpus of programs written in C, as well as the corresponding assembly produced by `gcc` with different levels of optimization (0 to 3), used for further training of pre-trained LLMs.
2. A hierarchical attention mechanism, structured to summarize the content of each instruction into the representation of a single token. This mechanism is compatible with existing models.
3. Two auxiliary contrastive loss objectives: a "functionality" one that minimizes the distance between representations of the same original code, while maximizing the distance between representations of different code pieces, and an "optimization" one encoding the fact that further levels of optimization should increase the distance between program representations.

Two variants (with 1B and 6B parameters respectively) of a model trained with these changes, and further fine-tuned for the task of interest, show a large improvement over state-of-the-art.

**Strengths:**

Originality
--------------
1. While hierarchical attention mechanisms are not new, the design of this one is innovative in that: it takes into account the specific format and constraints of assembly instructions, and it accommodates for using regular tokens in the same sequence (e.g., natural text instructions).
2. The contrastive objective losses, as well, encode a priori knowledge of the underlying data: compilation stages preserve semantics, and optimization stages are sequential.

Quality
----------
The different contributions are overall sensible, and contribute to the performance of the model. Experiments are adequately designed, and support the conclusions of the paper. The additional experiments help understand the role of the different contributions, in particular their effect on how embeddings get clustered and the effect it can have on the final model's performance.

Clarity
---------
1. The paper includes most of the relevant information, either in the main text or appendix. Relevant literature is mentioned and cited.
2. Figures and examples make it much easier to understand the logic, especially Fig. 3.

Significance
-----------------
1. This work shows a significant improvement on benchmarks, sustained across model sizes, and adaptable to other models. This is an advancement on an important, developing field of machine learning applications.
2. Given that these improvements do not require any in-depth change (e.g., to the vocabulary) and are compatible with already pre-trained model make it easier to experiment with in different settings.

**Weaknesses:**

Quality
----------
1. One of the 3 motivating cases in the introduction, malware detection, is not evaluated or considered at all in the rest of the paper. I understand the scope of the paper needs to end somewhere, but it would have strengthened the paper to include experiments on such a dataset.
2. Details are missing in how the authors are certain that test data sets (both for decompilation and for similarity detection) do not overlap with any of the training data, including the pre-training data of DeepSeek-Coder, even inadvertently.
3. An additional ablation on $\textrm{Nova}_{-CL}$ would have helped see if there are any non-linear interactions between HA and CL.

Clarity
---------
The overall organization of the paper could be improved. Many times, a concept, method, or setting is used in context before being formally explained. For instance:
1. If the "Related Work" section is positioned earlier, it would help introduce the baseline models (DeepSeekCoder, LLM4Decompile) that are used in the previous "Results" section, as well as attention mechanisms, including LongCoder's, also used earlier.
2. When describing the new datasets, it should be clear much earlier that "source code" really means "C code" (in the caption of Table 1, for instance), "assembly" is X86 assembly (or maybe X86-64? that's not so clear), that only `gcc` is considered as a compiler, and whether each "program" actually means a full executable program, or if it includes functions as well.
3. Similarly, the contrastive losses mention "the embedding" of a function, which is quite ambiguous in transformers, especially if the model family (encoder-decoder?) is not mentioned.
4. There is also a lot of ambiguity in notation, or the semantics of different objects. For instance:
    * Do Table 1, and Appendix A.2, refer to the original "AnghaBench" and "The-Stack" datasets, or the new datasets constructed by the authors in Section 2.1? Maybe it would be better to name the new ones.
    * In Functionality CL, l. 208 says it "optimizes Nova with the constraint", but a constraint is not a loss or objective. l. 215, "constraints can be trained" do not really make sense. It's also not obvious how the loss defined at l. 220 actually implements (a relaxation of) these constraints. It's also not explained if the sum over $f_i \in F$ is actually done over all the million embeddings in the corpus, or how it's implemented in practice.
    * K is introduced in Section 2.5, and then in 3.3., but we don't know what kinds of values will be used in practice. Also, Table 2 uses "Pass@K", but that's not the same K.
    * In captions of Fig. 4 (b) and (d), the tables are more "designs" than "implementations"
    * In Fig. 4 (b), the 1-4 indices are unfortunate as, for instance, $O0_3$ reads a lot like `-oO3`
    * The equations at l. 220 and l. 266 have a really similar form, but the use of indices $i$ and $j$ is swapped between the two, making it a bit harder

Significance
-----------------
The results are somewhat limited by the use of a single assembly language, and a single compiler, but this is acknowledged and does not seem like a fundamental limitation.

Minor points
-----------------
l. 461: "cauclated" -> "calculated"?

In the bibliography:
- Vaswani et al. is actually from 2017, not 2023 (though the arXiv version has had an inconsequential update in 2023), and a venue should be indicated (I'd suggest NeurIPS rather than arXiv)
- Other articles are missing a venue or source
- Several articles have incorrect capitalization in the title due to the lack of curly braces, e.g., use `{CodeT5}` to avoid it being rendered as "Codet5".

**Questions:**

1. Why do the evaluation for code similarity detection use cosine similarity (l. 321) when the objective (l. 212) uses the l2 distance?
2. What is the underlying metric for the Pass@k in the decompilation evaluation? Exact match, or some more lenient equivalent? It seems wrong to use exact match when, for instance, variable names would be arbitrary.
3. In Table 4, the second row is exactly the "Nova-1B" row of Table 2, but I was under the impression that "Nova-1B" was more than just "DeepSeekCoder + Nova's attention", in particular the additional training data, and CL objective. Are the numbers off, or the caption, or did I miss something?
4. When creating the assembly datasets (Appendix A.1), why go all the way to compiling executables, then using `objdump` for disassembling, with the associated possibilities of failure, rather than dump the assembly in the first place with `gcc -S`?
5. Do you have preliminary results, citations, or intuition behind the "normalizing" step of the assembly language performed in Fig. 6, in particular the addition of spaces? Is that necessary?

Minor points:
1. In the numerator on l. 265, is $f_j^q$ supposed to be $f^q$? or $f_j^p$ for which the substitution wouldn't apply?
2. l. 300, how many samples do the GPT models perform, then, to be able to compute the Pass@10 in Table 2?

Edit after discussion and update
--------------------------------------------
The overall clarity has improved, and additional information was provided.
Most questions have been answered, so I'm raising my score.

---

> ### Author Response · Authors · 2024-11-22
>
> We thank the reviewer for the insightful review and questions.
>
> ### 1. Clarification
>
> * The related work section has been moved to follow the introduction section.
> * We have highlighted in the data collection section that Nova focuses on C source code and X86-64 Assembly code.
> * We appreciate the careful review and suggestions. We revised the text accordingly and tried to solve the ambiguity (notations, citations, etc.).
>
> ### 2. L2 Distance versus Cosince SImilarity
>
> The reason we use L2 distance during training is that LLMs (e.g., DeepSeek-Coder on which we build Nova) hidden states are not normalized. Thus, the embedding we obtained for each source code and assembly function is not normalized. So we use L2 distance instead of adding normalization for simplicity.
>
> During the evaluation of the BCSD task. We reuse the framework provided in the baseline work CodeArt, which uses cosine similarity to rank the assembly functions. We normalize Nova’s embedding of assembly functions during the evaluation, and we know **the L2 distance after normalization keeps the same order as cosine similarity (smaller L2 distance means higher cosine similarity)**. So using normalization and cosine similarity during evaluation will not affect Nova’s performance.
>
> ### 3. Pass@K
>
> Pass@K is the most popular metric used to evaluate the correctness of code generation, defined as Equation 1 in the Codex’s paper [1].
>
> To be brief, Pass@K measures **how often** the model produces functional correct code. Higher Pass@K means we are more likely to see correct decompilation if we let the model sample K decompilations.
>
> ### 4. Table 4 Clarification
>
> In Table 4, the “DeepSeekCoder + Nova’s Attention” is essentially Nova-1B, and “DeepSeekCoder + LongCoder’s Attention” is Nova-1B but replacing our hierarchical attention with LongCoder’s Attention. The rest are the same: both models are trained with the same data, and the same procedure (including CL). We clarify this in the updated version.
>
> ### 5. Using `objdump -S` or `gcc -S`
> `gcc -S` is an alternative way of collecting the Assembly data, and Nova can be generalized to the Assembly produced by using `gcc -S`.
>
> However, a key difference is that the assembly generated by `gcc -S` does not undergo the linking step. In practical scenarios, binary decompilation and analysis are typically performed on executable or linked assembly code, as it includes linker modifications and reflects the final binary structure.
>
> Besides, our data collection aligns with many existing baselines, such as LLM4Decompile and jTrans, that collect their assembly code from executable binaries.
>
> ### 6. Normalization
>
> * Existing code LLMs’ tokenizers do not fit the assembly code very well. For instance: `’(%rax),%rbx’` is tokenized to `‘(‘ ‘%’ ‘ra’ ‘x’ ‘),’ ‘%’ ‘rb’ ‘x’`, where `’),’` is considered as one token. This does not properly show the semantics, since in X86 Assembly, parentheses are explicitly used to denote memory addressing while commas are used to separate operands. As we cannot re-train the tokenizer, we normalize the assembly code to make the tokenization better reflect assembly code semantics. We add white space to separate `‘(‘`, `‘)’` and `‘,’`, so that they are always considered as separate tokens.
> * We remove all the hexadecimal values as LLMs are very poor at understanding hexadecimal values.
> * Instead, we use the `[INST-i]` tokens to index the instructions. In decoder-only generative LLM, we want them to summarize the semantics of each instruction and thus have to put them at the end of each instruction since decoder-only LLM only has single-direction attention.
>
> ### 7. Clarification on Typo in $L_{BCSD}$
>
> Yes, thank you for pointing out the typo. The numerator $f^q_j$ should be $f^p_j$, $f^p_j$ is the positive candidate in the pool that comes from the same source code function as the query $f^q$. We have fixed it in the updated version.
>
> ### 8. Clarification on GPT Inference
>
> The GPT models also sample 20 decompilation per Assembly code using the same hyper-parameter (temperature 0.2, top-p 0.95), we rephrase the text to make it clearer.
>
> Reference:
> [1] https://arxiv.org/abs/2107.03374

---

> > ### Comment · Reviewer_MUPt · 2024-11-26
> >
> > 1. Clarification. Thanks for the updates to the paper, I believe it's a good improvement to clarity. However, some of my points in "Clarity 4." still stand (e.g., one does not train or optimize a constraint).
> > 2. If you're re-using an existing framework and methodology, I think it should be explicitly mentioned. The information about normalization at evaluation time, and your point about order preserving, should also be present (maybe in the appendix).
> > 3. Here as well, using the unbiased estimator of Codex for computing pass@k should be mentioned in the paper. But my question was more about what was considered "correct": are you relying on correct execution of unit tests, as in the Codex paper?
> > 4. Thanks for updating the table and text. But wouldn't "DeepSeekCoder + Nova's attention" be closer to "Nova_{-CL}"? Isn't the contrastive learning important?
> > 5. That makes sense, but maybe mention it in the appendix as well.
> > 6. Normalization
> >   * I understand the intuition, that makes sense. The LLMs may be capable to generalize a bit and understand that some tokens represent semantics from two successive things, but there's no real harm in making it explicit.
> >   * By "remove" here, do you mean "convert to decimal" as in l. 167? it seems that way in Figure 6.
> > 7. and 8. OK, thank you

---

> ### Author Response · Authors · 2024-11-27
>
> We upload a new version of the PDF.
>
> 1. "Train constraints" is indeed confusing. We revise the text between Line 260--268. We means "train Nova so that its embeddings for functions satisfy the constraints". Line 275--277 explains that in practice, the loss is calculated among each batch of functions (F is a batch of functions, not the entire dataset)
> 2. This is added to the Appendix A.6
> 3. Yes, we run test cases to validate correctness and calculate Pass@K. This is added in line 355--356.
> 4. I think our initial naming of "DeepSeekCoder + Nova's attention" confuses you. The reason you find that the numbers in the second row in Table 4 is the same as Nova-1B in Table 2 is because they are indeed the same model. We are comparing "DeepSeekCoder + LongCoder Attention + CL" with "DeepSeekCoder + Nova Attention + CL" in Table 4. The only difference between the two rows is replacing Nova's attention from Nova-1B with LongCoder's attention.
> 5. This is added to the Appendix in Line 892--899.
> 6. I agree with your point. This design is more by intuition before running experiments, and we do not have enough resources to test on this design choices. Yes, LLMs may be capable to generalize directly.
> By removing, I mean removing the address/offset of each instruction, e.g., for instruction "4: push %rbp", "4" is the address/offset. We remove this and use `[INST-i]` to indexing each instructions. For other hexadecimal values used in instructions, we convert them to decimal values.
>
> Let me know if you have additional concerns, we still have time before the end of Nov. 27th to update the PDF. Thank you.

---

### Author Response · Authors · 2024-11-22

We sincerely thank all the reviewers for their insightful review and questions. We have uploaded an updated version of our paper. The modifications are marked in blue:
* We moved the related work section to follow the introduction section to provide the reader with the background earlier.
* We revised the data collection section.
* We added clarification about the embeddings, and how the embeddings are obtained from the Nova model.
* We revised the text about the evaluation of the binary code decompilation task.
* We provided the study of the impact of each contrastive learning objectives on the binary cod decompilation task in the Appendix.

---

### Meta-Review · Area_Chair_pNVS · 2024-12-21

**Metareview:**

This paper studies using LLMs for binary code analysis. Despite many works used LLMs on source code tasks, they cannot be directly generalized to assembly code. The authors proposed an attention mechanism to capture the semantics in binary code for LLM training. Their method can outperform latest binary code similarity detection.

After rebuttal, this paper received diverse scores including three positive scores and two negative ones. I read all comments and rebuttals, especially Reviewer hTSw who gave a score of 3. Despite there is no reply from Reviewer hTSw, I think almost all questions are addressed. I think this paper can be accepted.

**Additional Comments On Reviewer Discussion:**

There are five reviews on this paper but only one reviewer participate the rebuttal (simple reply without any real discussion). After trying to ping them, I need to decide whether the questions are solved in rebuttal especially for those two negative reviews. The reviewer hTSw gave a lot of detailed questions of this paper but did not propose any key issues that can directly lead to a reject. The rebuttal is clear and all questions are answered. So even without confirmation from reviewers, I feel this paper should not be rejected.

---

### Decision · Program_Chairs · 2025-01-22

Accept (Poster)